# TSGGuide: Recommendation Guide for Multivariate Time Series Generation

## Abstract

Multivariate Time Series Generation (MTSG) plays a crucial role in time series analysis, supporting tasks such as data augmentation and anomaly detection. While several methods exist for MTSG, recommending the most suitable method for new scenarios remains a significant challenge. Although prior work by (Ang et al., 2023a) provides guidance for selecting MTSG methods, it lacks coverage of recent diffusion-based methods and has limited exploration of channel-independent frameworks. We address these gaps by improving the recommendation guide, highlighting the effectiveness of a central discriminator within the channel-independent framework. Our revised guide makes three key recommendations: 1) VAE-based methods excel on small-scale datasets; 2) a channel-independent framework with the newly designed central discriminator is optimal in most cases; and 3) a diffusion-based method is preferable when ample data and computational resources are available.

## 1 Introduction

Time series generation (TSG) involves producing synthetic sequences of temporally ordered data that mimic the statistical properties of real-world time series. Multivariate time series generation (MTSG) has gained prominence due to its applications in tasks like data augmentation (Ramponi et al., 2018), time series prediction (Cirstea et al., 2022; Wu et al., 2021), and anomaly detection (Ang et al., 2023b; Campos et al., 2021).

Numerous methods have been developed for MTSG (Ang et al., 2023a), typically falling into two categories: 1) channel-mixing frameworks, which merge time series features using models like generative adversarial networks (GANs) (Donahue et al., 2018; Yoon et al., 2019; Smith & Smith, 2020; Lin et al., 2020; Liao et al., 2020), variable autoencoders (VAEs) (Desai et al., 2021; Lee et al., 2023; Li et al., 2023; Naiman et al., 2024), flow-based methods (Chen et al., 2018; Kidger et al., 2021; Rubanova et al., 2019; Jeon et al., 2022; Zhou et al., 2023), and diffusion-based methods (Kong et al., 2021; Huang et al., 2024; Coletta et al., 2023; Yuan & Qiao, 2024); and 2) channel-independent frameworks with a central discriminator (Seyfi et al., 2022), which independently capture single-channel information while the discriminator handles inter-channel correlations.

Choosing an appropriate TSG method depends on domain-specific factors such as data volume, periodicity, and trends. For instance, Fourier flow (Alaa et al., 2021) is well-suited for tasks involving autocorrelation, while TimeVAE (Desai et al., 2021) performs better on smaller datasets. Therefore, a robust recommendation guide is essential for users. However, existing analyses, like (Ang et al., 2023a), may be incomplete—favoring channel-mixing frameworks, particularly VAE-based methods, while neglecting advanced diffusion models and offering limited exploration of channel-independent frameworks. For example, the periodicity of the multivariate time series and the correlation between the channels are important features. However, the central discriminator designed in (Seyfi et al., 2022) does not adequately address this problem. In addition, (Ang et al., 2023a) has not tested the newly proposed diffusion model-based methods and needs to be upgraded to form a reliable recommendation guide for time series generation (Kong et al., 2021; Huang et al., 2024; Coletta et al., 2023; Yuan & Qiao, 2024).

In light of the insufficient comparison with advanced diffusion-based generative frameworks and limited exploration of channel-independent frameworks, we have updated the recommendation guidelines from (Ang et al., 2023a) to better aid users in selecting TSG frameworks. Specifically, we

advocate for an increased priority recommendation for both channel-independent and diffusion-based generative frameworks, forming an enhanced TSGGuide.

## 2 RELATED WORK

### 2.1 CHANNEL MIXING-BASED MTSG

There is a predominant focus on channel-mixing time series generation framework. Some methods (Mogren, 2016; Esteban et al., 2017; Wiese et al., 2020; Xu et al., 2020; Jarrett et al., 2021; Jeha et al., 2021) employ GAN-based (Goodfellow et al., 2014) architecture combined with neural networks like LSTM (Hochreiter & Schmidhuber, 1997), GRU (Cho et al., 2014), and Transformer (Vaswani et al., 2017), which excel at capturing sequential data. Certain methods leverage VAEs (Kingma & Welling, 2013) to effectively capture temporal features through variational inference, resulting in efficient models with potential interpretability. Recent advancements have explored hybrid methods, combining flow-based models with techniques such as ODE (Chen et al., 2018), or integrating them with GANs or VAEs. Moreover, several diffusion-based methods (Yuan & Qiao, 2024; Kong et al., 2021; Coletta et al., 2023; Huang et al., 2024) have been designed to handle MTSG. Due to their stable training and high-quality generation, diffusion models exhibit promising performance on numerous datasets. Employing channel-mixing framework intuitively allows for improved consideration of correlations between channels in multivariate time series data, thereby enhancing model performance.

### 2.2 CHANNEL INDEPENDENCE-BASED MTSG

COSCIGAN (Seyfi et al., 2022) employs channel GANs and a central discriminator to generate multivariate time series, but it fails to effectively capture key characteristics like periodicity and inter-channel correlations, limiting the performance of the central discriminator. Compared to the MLP-based discriminator in COSCIGAN, more advanced architectures such as attention mechanisms, TimesNet (Wu et al., 2022), and ModernTCN (donghao & wang xue, 2024) offer better potential for performance improvement. Our findings suggest that the central discriminator operates primarily as a time series classifier in small-sample contexts. However, MLP-based methods, as well as attention-based approaches like TimesNet, exhibit limited effectiveness in handling such scenarios.

## 3 BACKGROUND

### 3.1 PROBLEM DEFINITION

**Dataset Setup.** Suppose that an MTS has length $L$ with $N$ channels. The general length of these time series is often long, making it difficult to input the entire time series data at once and extract its features. To generate time series within a short period and extract features from the time series data, we need to choose an appropriate subsequence length, denoted as $l$, and use a step size of 1 when performing the partition. Then we transform MTS into $\mathbf{T} \in \mathbb{R}^{K \times l \times N}$, where $K = L - l + 1$. Additionally, we normalize the dataset to the range of $[0, 1]$ to enhance efficiency and numerical stability.

**Time Series Generation.** We define $p(\mathbf{T})$ as the true distribution of the time series. Our goal is to create a synthetic time series $\widehat{\mathbf{T}} \in \mathbb{R}^{K \times l \times N}$ in which its distribution $q(\widehat{\mathbf{T}})$ is similar to $p(\mathbf{T})$.

### 3.2 CHANNEL MIXING-BASED MTSG

Channel-mixing frameworks in time series generation models aim to learn the joint distribution of all channels in a multivariate time series, regardless of the number of channels. The goal of these frameworks can be defined as follows: $\min_{q} D(p(\mathbf{T}) \| q(\widehat{\mathbf{T}}))$, where $D$ is any suitable measure of the distance between two distributions. Irrespective of the utilization of GANs, VAEs, or flow-based methods, most of them employing a channel-mixing framework strive to learn an improved $q(\widehat{\mathbf{T}})$.

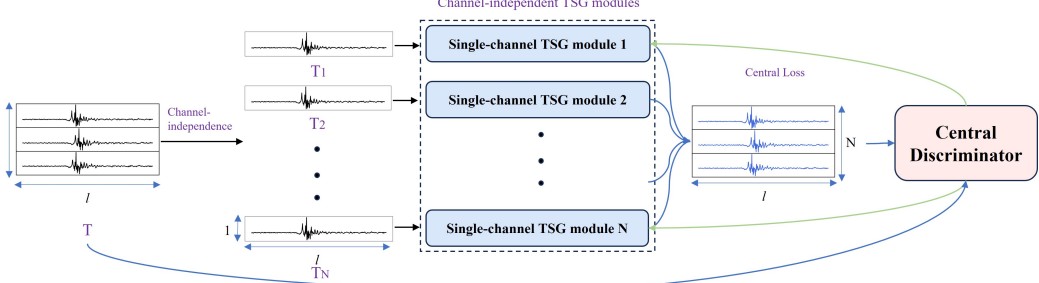

Figure 1: Structure of channel independece-based MTSG.

### 3.3 Channel Independence-based MTSG

Figure 1 illustrates the structure of the channel-independent framework with a central discriminator, comprising two main components: 1) Channel-independent TSGs; 2) Central discriminator.

**Channel-independent TSGs**  Here, each single-channel TSG module synthesizes one single-channel time series. The input $\mathbf{T}$ is divided into $N$ single-variable sequences $\boldsymbol{T}_i \in \mathbb{R}^{K \times l \times 1}$, where $i \in [1, N]$. Subsequently, following the channel-independent setup, the $i$-th channel's time series $\boldsymbol{T}_i$ is input into the $i$-th single-channel TSG module, yielding the generated single-channel time series data $\widehat{\boldsymbol{T}}_i = G_{i,\theta_i}(z)$, where $G$ is a TSG module and $\theta_i$ is the parameters of the $i$th module. This framework allows flexibility in selecting channel-independent generators, such as GANs and VAEs.

In the channel objective, we aim to find a distribution $q(\widehat{\boldsymbol{T}}_i)$ that closely approximates the true channel distribution $p(\boldsymbol{T}_i)$ of the dataset. Subsequently, by concatenating $N$ instances of $\widehat{\boldsymbol{T}}_i$, where $i = 1, \cdots, N$, along the second dimension, the generated time series $\widehat{\mathbf{T}}$ is obtained. For each channel $\boldsymbol{T}_i$, we optimize $\min_{q} D(p(\boldsymbol{T}_i) \| q(\widehat{\boldsymbol{T}}_i))$. When channel-independent generators are trained, incorporating central discriminator loss suffices.

**Central Discriminator**  In the center objective, we consider the overall characteristics of multivariate time series, which involves integrating the time series signals from all channels. This part designs powerful binary classifiers to distinguish between $\widehat{\mathbf{T}}$ and $\mathbf{T}$. Certainly, the more difficult it is to distinguish between $\widehat{\mathbf{T}}$ and $\mathbf{T}$ under this classifier, the better. We should optimize $\min_{q} D(p(\mathbf{T}) \| q(\widehat{\mathbf{T}}))$.

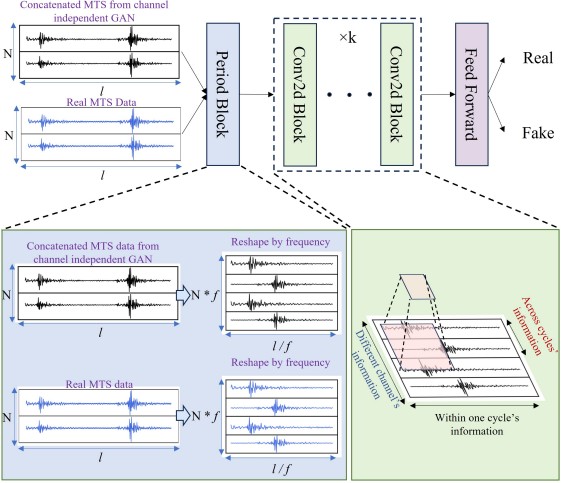

### 3.4 The Designed Central Convolution Discriminator

CCD uses Fast Fourier Transform (FFT) or Wavelet Transform (WT) to perform periodic segmentation of the samples, reducing

Figure 2: The structure of CCD.

the classification difficulty while revealing inter-period relationships within the samples to enhance key features that are beneficial for classification. It then uses 2D convolution to capture both intra-channel and inter-channel relationships. The details of CCD are shown in Appendix A. The deisigned CCD is shown in Figure 2.

**Period Block**  For a multivariate time series $T$, we utilize Frequency to obtain its periodicity in the frequency domain.

$$\boldsymbol{a} = Avg\left(Amp\left(Frequency(\mathbf{T})\right)\right) \tag{1}$$

where $Frequency(\cdot)$. represents the solution to obtain the frequency. Here, we use FFT or WT. $\boldsymbol{a}$ represents the calculated amplitude of each frequency, which is averaged from the dimensions $N$ using $Avg(\cdot)$.

The true periodicity of $\mathsf{T}$ is computed using $l_p = \lceil \frac{l}{f} \rceil$. Depending on the selected frequency and the corresponding period length, we can transform the dimension of the multivariate time series $\mathsf{T}$ and $\widehat{\mathsf{T}}$ into $(K, l_p, N \times f)$.

$$\mathsf{T}_p \in \mathbb{R}^{K \times l_p \times f}, \widehat{\mathsf{T}}_p \in \mathbb{R}^{K \times l_p \times f} = Reshape(\mathsf{T}, \widehat{\mathsf{T}}) \qquad (2)$$

**Conv2d Block**   After passing through the Period module, we utilize multiple conv2d blocks to capture the three types of local information mentioned earlier, distinguishing between real and synthesized multivariate time series. The formulation is as follows:

$$\boldsymbol{T}_c, \widehat{\boldsymbol{T}_c} = Conv2d\ Block(\mathsf{T}_p, \widehat{\mathsf{T}}_p) \qquad (3)$$

where we transform 2D representations $\mathsf{T}_p, \widehat{\mathsf{T}}_p \in \mathbb{R}^{K \times l_p \times f}$ into 1D space $\boldsymbol{T}_c, \widehat{\boldsymbol{T}}_c \in \mathbb{R}^{K \times d}$. Considering both performance and efficiency, we opt for experiments using the nn.conv2d() block based on PyTorch for our main experiments.

Finally, the features extracted through the Conv2d blocks are further processed by the feedforward module, which consists of a linear layer followed by a sigmoid activation function, yielding the ultimate classification results.

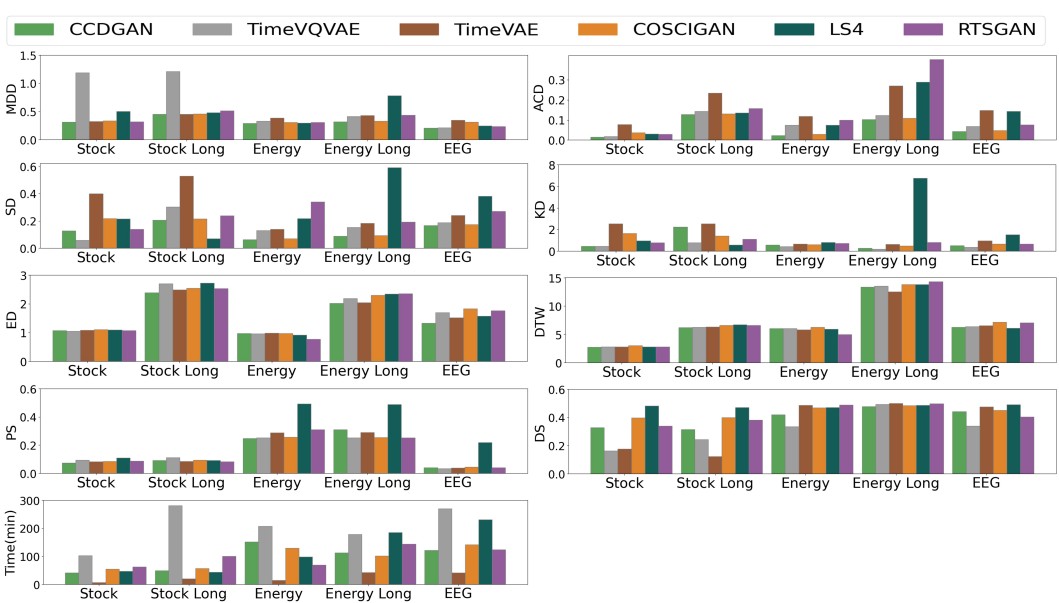

Figure 3: Results of six methods on real datasets.

# 4 EXPERIMENTS

## 4.1 VALIDATION OF TSGGUIDE

This section explores the performance of current typical time series generation models on different domain data and forms a user-oriented guide for recommending time series generation models.

**Baselines**   According to the analysis provided by (Ang et al., 2023a), 1) channel-independent methods: COSCI-GAN (Seyfi et al., 2022); 2) channel-mixing methods: TimeVQVAE (Lee et al., 2023), TimeVAE (Desai et al., 2021), RTSGAN (Pei et al., 2021), and LS4 (Zhou et al., 2023) generally outperform other time series generation models. Regarding the experimental design, to ensure fairness, the channel GANs in CCD employ LSTM models. The detailed information on parameters metrics and datasets can be found in Appendix Section C, D, and E.

**Parameters** For RTSGAN, we adhere to its complete time series generation (Pei et al., 2021) and set $\beta_1 = 0.9$ and $\beta_2 = 0.999$. For COSCI-GAN, we set $\gamma = 5$, employ MLP-based networks for the central discriminator, and follow other hyper-parameters from (Seyfi et al., 2022). For TimeVAE, we set the latent dimension to 8 and the hidden layer sizes to 50, 100, and 200. For TimeVQVAE, we adopt the settings from (Lee et al., 2023), with $n\_fft = 8$ and varying $max\_epochs \in \{2000, 10000\}$ for two training stages. For LS4, we set the latent space dimension to 5 and configured the batch size to 1024.

**Datasets** To validate the performance of CCDGAN in real multivariate time series and assess its generality, we selected five datasets: Stock (Yoon et al., 2019), Stock Long (Yoon et al., 2019), Energy (Candanedo, 2017), Energy Long (Candanedo, 2017), EEG (Roesler, 2013), DLG (Hutchins, 2006), and Air (Zheng et al., 2015), covering the finance, energy, traffic, air, and medical domains (see Appendix C for more information). We focus on datasets with a large number of channels.

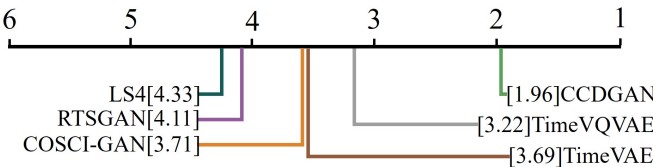

Figure 4: The critical difference diagram illustrates the performance ranking of six algorithms across five datasets employing Wilcoxon-Holm analysis (Ismail Fawaz et al., 2019) at a significance level of $p = 0.05$. Algorithm positions are indicative of their mean ranks across multiple datasets, with higher ranks signifying a method consistently outperforming competitors. Thick horizontal lines denote scenarios where there is no statistically significant difference in algorithm performance.

**Results** The results shown in Figures 4 and 3 provide evidence for the effectiveness of CCDGAN in synthesizing MTS data. We also conducted experiments based on DLG and Air, which are from the traffic flow dataset and air dataset respectively. For more dataset details, see Appendix C. The experimental results are shown in Tables 1 and 2. The performance of CCDGAN surpasses that of COSCI-GAN, validating the effectiveness of the proposed CCD module. As shown in Table 11, we choose the LSTM, which exhibits moderate generation performance, as the channel-independent generator. However, its performance exceeds that of the current state-of-the-art channel-mixing methods, TimeVQVAE and TimeVAE. Particularly on datasets with a higher number of channels such as Energy and Energy Long, the superiority of CCDGAN is more pronounced. Relative to COSCI-GAN, the proposed framework does not consume additional time. Furthermore, in comparison to channel-mixing methods, the runtime of the channel-independent approach is competitive.

Table 1: Results on DLG dataset.

| Method | CCDGAN | TimeVQVAE | TimeVAE | COSCI-GAN | LS4 | RTSGAN |
|--------|--------|-----------|---------|-----------|-----|--------|
| MDD↓ | 0.284 | 0.293 | 0.301 | 0.241 | **0.227** | 0.238 |
| ACD↓ | 0.121 | 0.162 | 0.164 | 0.137 | **0.117** | 0.178 |
| SD↓ | 0.214 | 0.235 | **0.209** | 0.257 | 0.216 | 0.227 |
| KD↓ | 12.341 | 12.25 | **12.175** | 12.275 | 12.377 | 12.375 |
| ED↓ | **1.12** | 1.363 | 1.315 | 1.641 | 1.363 | 1.177 |
| DTW↓ | 2.346 | 2.367 | 2.476 | **2.316** | 2.375 | 2.438 |
| PS↓ | **0.415** | 0.488 | 0.446 | 0.461 | 0.425 | 0.427 |
| DS↓ | 0.245 | **0.221** | 0.237 | 0.257 | 0.227 | 0.288 |

**Results on Small Data** We simulate the scenario of limited data and assess the model's generalization ability on small data. To ensure an accurate evaluation, we randomly selected 10% and 20% of the Stock dataset and utilized these subsets for model training, and then the entire dataset was employed for evaluation. We also tested on 10% Energy and 10% EEG data and the results can

Table 2: Results on Air dataset.

| Method | CCDGAN | TimeVQVAE | TimeVAE | COSCI-GAN | LS4 | RTSGAN |
|---|---|---|---|---|---|---|
| MDD↓ | 0.139 | 0.127 | 0.121 | 0.142 | 0.148 | **0.117** |
| ACD↓ | **0.109** | 0.117 | **0.109** | 0.117 | 0.115 | 0.126 |
| SD↓ | 0.356 | **0.328** | 0.387 | 0.382 | 0.361 | 0.371 |
| KD↓ | **8.147** | 8.278 | 8.178 | 8.187 | 8.169 | 8.171 |
| ED↓ | **0.816** | 0.828 | 0.827 | 0.826 | 0.829 | 0.877 |
| DTW↓ | 2.044 | 2.28 | 2.091 | 2.081 | 2.062 | **2.027** |
| PS↓ | **0.404** | 0.483 | 0.426 | 0.433 | 0.429 | 0.416 |
| DS↓ | **0.107** | 0.186 | 0.131 | 0.124 | 0.139 | 0.121 |

Table 3: Results on 10% and 20% Stock Dataset.

| 10% Stock | CCDGAN | TimeVQVAE | TimeVAE | COSCI-GAN | LS4 | RTSGAN |
|---|---|---|---|---|---|---|
| MDD↓ | 0.511 | 3.092 | **0.472** | 0.606 | 1.524 | 1.489 |
| ACD↓ | 0.049 | **0.027** | 0.173 | 0.083 | 0.197 | 0.203 |
| SD↓ | 0.2 | **0.133** | 0.751 | 0.581 | 0.862 | 0.467 |
| KD↓ | 0.861 | 1.492 | 1.023 | 1.422 | 1.013 | **0.799** |
| ED↓ | 2.239 | 1.98 | **1.442** | 1.935 | 1.928 | 1.842 |
| DTW↓ | 5.717 | 5.64 | 4.952 | 4.913 | 4.898 | **4.645** |
| 20% Stock | CCDGAN | TimeVQVAE | TimeVAE | COSCI-GAN | LS4 | RTSGAN |
| MDD↓ | 0.49 | 2.982 | **0.443** | 0.582 | 1.398 | 1.207 |
| ACD↓ | 0.438 | 0.228 | **0.166** | 0.182 | 0.231 | 0.192 |
| SD↓ | **0.088** | 0.127 | 0.739 | 0.428 | 0.814 | 0.308 |
| KD↓ | 0.85 | 1.321 | 1.006 | 1.391 | 0.838 | **0.799** |
| ED↓ | 2.099 | **1.089** | 1.382 | 1.752 | 1.79 | 1.661 |
| DTW↓ | 5.37 | 5.246 | 3.812 | 4.521 | 4.682 | **4.568** |

be seen in Appendix Tables 18 and 19. The results in Table 3 clearly demonstrate that TimeVAE outperforms other methods on small datasets.

This observation can be attributed to two factors: (1) The smaller dataset size reduces distributional complexity, amplifying inter-channel correlations, and (2) models utilizing channel-mixing are better equipped to capture these correlations, as they process data across multiple channels simultaneously, unlike channel-independent methods. Additionally, VAE-based methods demonstrate superior performance, likely due to their stable training dynamics, which demand less data.

**Results on Simulated Data**   We evaluated the model's robustness and overall performance on simulated datasets, complementing the results obtained from real-world datasets. The simulated data generation process follows the method outlined in Meidani et al. (2023) (Section 3.3), with detailed

Table 4: Results on Simulated Data

| Method | CCDGAN | TimeVQVAE | COSCI-GAN | LS4 | Diffusion-TS |
|---|---|---|---|---|---|
| MDD↓ | 0.251 | 0.282 | 0.272 | 0.258 | **0.247** |
| ACD↓ | 0.116 | 0.128 | 0.107 | **0.105** | 0.114 |
| SD↓ | **0.241** | 0.259 | 0.244 | 0.248 | 0.246 |
| KD↓ | **11.01** | 11.017 | 11.028 | 11.027 | 11.014 |
| ED↓ | 1.153 | 1.151 | 1.158 | 1.155 | **1.151** |
| DTW↓ | **2.466** | 2.469 | 2.474 | 2.471 | 2.472 |
| PS↓ | 0.164 | **0.162** | 0.169 | 0.175 | 0.166 |
| DS↓ | 0.205 | 0.217 | 0.212 | 0.219 | **0.202** |

implementation provided in Appendix L. Using 100 functions, we generated simulated time series datasets and applied the same experimental setup and evaluation metrics as in Figure 3. Alongside CCDGAN, we compared four other methods: TimeVQVAE, COSCIGAN, LS4, and DiffusionTS, each representing different generative frameworks and demonstrating strong performance on real datasets. The results are presented in Table 4.

The results demonstrate that both the diffusion-based channel-mixing framework Diffusion-TS and the GAN-based channel-independent framework CCDGAN excel across multiple evaluation metrics, with each achieving top rankings in three categories. CCDGAN outperforms in two distance-based metrics, likely due to GANs' strength in capturing temporal distance characteristics. Meanwhile, Diffusion-TS exhibits balanced performance across all metrics, highlighting the overall effectiveness of diffusion-based generation models.

**CCDGAN v.s. Diffusion-based Methods**    In recent years, diffusion models have been increasingly applied to MTSG, achieving notable results. From the above experimental results, CCDGAN performs better when the amount of data is sufficient. We replicate the experimental setup from (Yoon et al., 2019), detailed in Appendix Section I.

*Baselines*. To facilitate a fair comparison among different approaches and provide updated recommendations for MTSG methods, we compare CCDGAN with three diffusion-based TSG methods: Diffwave (Kong et al., 2021), DiffTime (Coletta et al., 2023), and Diffusion-TS (Yuan & Qiao, 2024). These methods, including Diffusion-TS, DiffWave, and DiffTime, are all based on DDPM, with modifications tailored to the characteristics of time series data.

*Parameters*. For Diffusion-TS, DiffWave and DiffTime, to ensure the fairness of the experiments, we strived to maintain consistency in parameter settings. We chose 4 attention heads, each with a dimension of 16, and selected 2 encoder and decoder layers.

*Datasets*. For dataset selection, we chose the previously mentioned Stock and Energy datasets. For the Sine dataset, we opted for the sine wave dataset provided in (Yoon et al., 2019), which is channel-independent and exhibits more diverse variations.

Table 5: Results between CCDGAN and Diffusion-based methods. Bold indicates best performance.

| Metric | Methods | Sines | Stocks | Energy |
|---|---|---|---|---|
| Context-FID↓ | CCDGAN | 0.008±.001 | **0.158±.022** | 0.134±.019 |
| | Diffusion-TS | **0.006±.000** | **0.147±.025** | **0.089±.024** |
| | Diffwave | 0.014±.002 | 0.232±.032 | 1.031±.131 |
| | DiffTime | 0.006±.001 | 0.236±.074 | 0.279±.045 |
| Correlational Score↓ | CCDGAN | **0.016±.000** | 0.023±.012 | **0.823±.108** |
| | Diffusion-TS | **0.015±.004** | **0.004±.001** | 0.856±.147 |
| | Diffwave | 0.022±.005 | 0.030±.020 | 5.001±.154 |
| | DiffTime | **0.017±.004** | 0.006±.002 | 1.158±.095 |
| Discriminative Score↓ | CCDGAN | **0.009±.000** | 0.138±.042 | 0.131±.000 |
| | Diffusion-TS | **0.006±.007** | **0.067±.015** | **0.122±.003** |
| | Diffwave | 0.017±.008 | 0.232±.061 | 0.493±.004 |
| | DiffTime | 0.013±.006 | 0.097±.016 | 0.445±.004 |
| Predictive Score↓ | CCDGAN | **0.093±.000** | 0.041±.000 | **0.250±.000** |
| | Diffusion-TS | **0.093±.000** | **0.036±.000** | **0.250±.000** |
| | Diffwave | **0.093±.000** | 0.047±.000 | 0.251±.000 |
| | DiffTime | **0.093±.000** | 0.038±.001 | 0.252±.000 |
| Training Time(min)↓ | CCDGAN | **11** | **10** | **37** |
| | Diffusion-TS | 17 | 15 | 60 |
| | Diffwave | 19 | 16 | 68 |
| | DiffTime | 19 | 16 | 69 |
| 2000 Data Sampling Time(s)↓ | CCDGAN | **5** | **7** | **13** |
| | Diffusion-TS | 23 | 26 | 65 |
| | Diffwave | 24 | 30 | 70 |
| | DiffTime | 24 | 29 | 69 |

Table 6: Results between CCDGAN and Diffusion-TS on 10% Stock Dataset. Bold indicates best performance.

| Method | CCDGAN | Diffusion-TS |
|--------|--------|--------------|
| MDD↓ | **0.511** | 0.613 |
| ACD↓ | **0.049** | 0.058 |
| SD↓ | **0.2** | 0.244 |
| KD↓ | **0.861** | 1.015 |
| ED↓ | 2.239 | **1.996** |
| DTW↓ | **5.717** | 5.931 |

The results, presented in Table 5, indicate that diffusion-based methods generally exhibit superior performance. However, they also require significant time and computational resources. For those prioritizing stable training and high-quality results, diffusion-based methods are recommended. Nonetheless, CCDGAN outperforms certain diffusion-based methods in terms of performance while demanding less time and computational resources, highlighting its competitive advantage in both effectiveness and efficiency.

We tested CCDGAN and DiffusionTS using a 10% Stock dataset to compare the performance of the two methods in small dataset scenarios. The hyperparameter settings are consistent with Table 3, and the experimental results are shown in Table 6. The test results show that CCDGAN performs better than Diffusion-TS on small datasets. This may be related to the inherent characteristics of the diffusion model. Since the training of the diffusion model usually requires a large amount of data, its performance under small dataset conditions is not as good as other generative models.

## 4.2 TSGGUIDE: RECOMMENDATION GUIDELINES

Finally, combining the guidelines from the study by TSGBench (Ang et al., 2023a), we offer guidance to assist users in effectively selecting suitable TSG methods. In contrast to TSGBench, the updated sections are shown in Appendix Section K.

1. In Figure 3, VAE-based methods demonstrate faster training times and rank above average on several metrics. This makes VAE-based methods suitable for initial attempts due to their lower time requirements. Additionally, the results in Tables 3, and Appendix Tables 18 and 19 show that VAE-based methods perform well on small datasets, making them applicable to various types of datasets. As a foundational step, we recommend that users start with the VAE-based method. Its consistently excellent computational efficiency makes it the preferred choice for initial exploration and handling of small data sets.

2. If we emphasize autocorrelation or forecasting, such as predictive maintenance or stock market analysis, the ACD measure becomes crucial. CCDGAN is highly suitable for these scenarios.

3. In Figure 3, CCD shows a leading performance on the majority of datasets, proving that CCDGAN performs well on real-world datasets, even though they come from different domains. When the dataset originates from a novel domain or exhibits complex multivariate relationships, CCDGAN is a recommended choice. It shows excellent performance across data sets from various domains and outperforms other methods, particularly on datasets with a large number of variables.

4. In Table 5, diffusion model-based methods achieve certain advantages in metrics. However, these methods require significant training and sampling time, leading to higher computational costs. Therefore, if ample computational resources are available, diffusion model-based approaches should be considered. When computational resources and dataset are abundant and optimal results are desired, it is recommended to utilize Diffusion-TS. These methods typically exhibit stable training and yield superior performance.

5. Users can further adjust method selection based on specific application requirements, including identifying the appropriate channel-independent TSG module (see Section J).

### 4.3 VALIDATION OF CCD

This section examines the effect of the central discriminator within the channel-independent generation framework. We compared COSCI-GAN using common attention mechanisms and TimesNet, a state-of-the-art time series classification framework, as baselines. Experiments were conducted on both the full Stock dataset and a reduced version with 10% of the data, maintaining the same number of channels in COSCI-GAN. The central discriminator in COSCI-GAN was replaced by the Transformer Encoder, TimesNet, and CCD for evaluation. The Transformer Encoder structure followed (Vaswani et al., 2017) with $N = 6$. Detailed information on parameter metrics and datasets is provided in Appendix Section C and D.

Table 7: The results of Transformer, Timesnet, COSCIGAN, and CCD on the Stock dataset and the 10% Stock dataset. Due to poor downstream task performance with 10% of the Stock data, comparative results are omitted. Bold numbers in the table indicate the best performance.

| | 100% Stock | | | | 10% Stock | | | |
|---|---|---|---|---|---|---|---|---|
| Methods | CCD | Transformer | Timesnet | COSCIGAN | CCD | Transformer | Timesnet | COSCIGAN |
| MDD↓ | **0.315** | 0.341 | 0.38 | 0.334 | **0.511** | 0.692 | 1.036 | 0.606 |
| ACD↓ | **0.015** | 0.027 | 0.029 | 0.037 | **0.049** | 0.132 | 0.135 | 0.083 |
| SD↓ | **0.127** | 0.149 | 0.133 | 0.217 | **0.200** | 0.431 | 0.344 | 0.581 |
| KD↓ | **0.446** | 0.449 | 0.485 | 1.647 | **0.861** | 1.392 | 1.492 | 1.422 |
| ED↓ | **1.068** | 1.094 | 1.100 | 1.101 | 2.239 | 2.58 | 2.98 | **1.935** |
| DTW↓ | **2.772** | 2.805 | 2.959 | 3.014 | 5.717 | 5.93 | 6.64 | **4.913** |
| PS↓ | **0.073** | 0.097 | 0.091 | 0.086 | - | - | - | - |
| DS↓ | **0.327** | 0.339 | 0.341 | 0.396 | - | - | - | - |

As shown in Table 7, these methods struggled with small sample scenarios, exhibiting a notable performance drop with 10% of the Stock data. The heavy parameterization of attention mechanisms likely made them less effective for limited data. TimesNet, designed for single-channel time series, failed to capture inter-channel dependencies, a critical aspect for multi-channel time series generation. In contrast, CCD demonstrated superior performance, particularly with the 10% Stock data, where its advantage was most pronounced. More discussion of CCD can be found in Appendix J.

## 5 CONCLUSION

In this work, we tackled the limitations in existing multivariate time series generation frameworks by proposing an updated recommendation guide, TSGGuide, aimed at improving the selection of generation methods. Our comprehensive analysis revealed gaps in prior work, particularly in the limited exploration of channel-independent frameworks and the absence of evaluations for diffusion-based models. We have demonstrated that by enhancing the central discriminator within the channel-independent framework—integrating methods like TimesNet and attention mechanisms—and ultimately introducing CCD, significant performance improvements can be achieved, particularly in small sample scenarios.

Our contributions offer a more balanced assessment of both channel-independent and diffusion-based methods, underscoring their value in time series generation tasks. This work provides a more nuanced framework for selecting TSG methods, addressing both the data-specific needs of users and the current state of the field. Future research should continue to investigate the evolving landscape of MTSG, with an emphasis on addressing periodicity and inter-channel correlation challenges. With the conditional time series generation problem posed, it makes sense to explore the advantages and disadvantages of these schemes.

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

# A    CCDGAN

Since time series generation tasks are motivated by small sample sizes, the central discriminator in the channel independent generation framework is essentially a small sample time series classifier. The main challenges in small-sample time series classification arise from the difficulty of adequately training neural networks with limited data, which reduces their classification performance. Possible solutions include: 1) performing data augmentation to increase the data size; 2) decomposing the problem to reduce the classification difficulty; 3) enhancing the information to capture key features that aid classification; and 4) using a small-parameter, high-performance mapping framework. The deisigned CCD is shown in Figure 2.

**Period Block**   In multivariate time series, changes in one channel often have short-term effects on other channels. COSCIGAN (Seyfi et al., 2022) may tend to overlook the crucial mechanism of sharing information between adjacent time steps. Additionally, considering the periodic nature of time series, models should be able to capture time patterns across different cycles. Each time point involves two types of temporal changes: **intra-cycle changes** and **inter-cycle changes**, which correspond to adjacent regions and different phases of the same cycle.

For a multivariate time series $T$, we utilize Frequency to obtain its periodicity in the frequency domain.

$$\boldsymbol{a} = Avg\left(Amp\left(Frequency(\mathbf{T})\right)\right) \tag{4}$$

where $Frequency(\cdot)$. represents the solution to obtain the frequency. Here, we use FFT or WT. $\boldsymbol{a}$ represents the calculated amplitude of each frequency, which is averaged from the dimensions $N$ using $Avg(\cdot)$.

In consideration of the sparsity in the frequency domain and to avoid noise caused by irrelevant high frequencies, we conducted testing and found that selecting only the first amplitude value is sufficient. We denote the non-normalized amplitude as $\boldsymbol{a}$ and obtain the most significant frequency $f$ from it. The true periodicity of $\mathbf{T}$ is computed using $l_p = \lceil \frac{l}{f} \rceil$. Depending on the selected frequency and the corresponding period length, we can transform the dimension of the multivariate time series $\mathbf{T}$ and $\widehat{\mathbf{T}}$ into $(K, l_p, N \times f)$.

$$\mathbf{T}_p \in \mathbb{R}^{K \times l_p \times f}, \widehat{\mathbf{T}_p} \in \mathbb{R}^{K \times l_p \times f} = Reshape(\mathbf{T}, \widehat{\mathbf{T}}) \tag{5}$$

It is worth noting that this transformation enhances three types of local information to the transformed 3D tensor: 1) local information between adjacent time points in the same channel (within one cycle), 2) local information between adjacent periods in the same channel (across cycles), and 3) local information of adjacent time steps in different channels. Therefore, through the receptive field of the convolution structure, CCD can better preserve the correlations between channels.

**Conv2d Block**   After passing through the Period module, we utilize multiple conv2d blocks to capture the three types of local information mentioned earlier, distinguishing between real and synthesized multivariate time series. The formulation is as follows:

$$\boldsymbol{T}_c, \widehat{\boldsymbol{T}_c} = Conv2d\,Block(\mathbf{T}_p, \widehat{\mathbf{T}_p}) \tag{6}$$

where we transform 2D representations $\mathbf{T}_p, \widehat{\mathbf{T}_p} \in \mathbb{R}^{K \times l_p \times f}$ into 1D space $\boldsymbol{T}_c, \widehat{\boldsymbol{T}_c} \in \mathbb{R}^{K \times d}$.

After conducting tests, we recommend adopting a progressive kernel size expansion strategy for convolution operations. The specific sizes for the kernel are $kernel\,size = 2 \times i + 1$, where $i = 1, 2, \cdots, k$ and $k$ is the number of Conv2d blocks. In addition to the aforementioned approach, we can choose various convolution modules from computer vision, such as widely used models like ResNet (He et al., 2016) and ResNeXt (Xie et al., 2017), for feature extraction within the Conv2d Block. Generally, stronger 2D backbones for representation learning lead to better performance. Considering both performance and efficiency, we opt for experiments using the nn.conv2d() block based on PyTorch for our main experiments. Later we will talk about the influence on kernel size.

Finally, the features extracted through the Conv2d blocks are further processed by the feedforward module, which consists of a linear layer followed by a sigmoid activation function, yielding the ultimate classification results.

**Loss Function**  In designing the loss function, our approach propagates the CD loss to each single-channel time series generation module. By incorporating CD loss terms, we aim to enhance the modules' focus on inter-channel correlations.

If we select pure GAN-based methods to be the single-channel time series generation module, the objectives function can be summarized as follows:

$$
\min_{\theta_i} \max_{\phi_i} \max_{\alpha} \mathcal{L} = \mathbb{E}_{\boldsymbol{T}_i \sim P_{\text{data}}}[\log(D_{i,\phi_i}(\boldsymbol{T}_i)) + \gamma \cdot \log(CCD_\alpha(\boldsymbol{T}_i))]
$$
$$
+ \mathbb{E}_{\boldsymbol{z} \sim P_{\boldsymbol{z}}}[\log(1 - D_{i,\phi_i}(G_{i,\theta_i}(\boldsymbol{z}))) + \gamma \cdot \log(1 - CCD_\alpha(G_{i,\theta_i}(\boldsymbol{z}), G_{j \neq i}(\boldsymbol{z})))]
$$
(7)

where $G_{i,\theta_i}$ represents the generator of the $i$-th channel independent GAN with parameters $\theta_i$, and $D_{i,\phi_i}$ represents the discriminator of the $i$-th channel independent generator with parameters $\phi_i$. $CCD_\alpha$ represents the central discriminator with parameter $\alpha$. $P_{\text{data}}$ represents the distribution of real time series. $G_{j \neq i}$ represents all other generators with fixed parameters, except $G_{i,\theta_i}$, in the optimization steps. $\gamma$ is a hyper-parameter that controls the balance between three types of locality among channels and generates higher-quality signals within each channel. $\boldsymbol{z}$ is a shared noise vector sampled from the distribution $P_{\boldsymbol{z}}$.

**Model Training**  Our training approach using the channel-independent framework is similar to that of COSCIGANs. In each training iteration, $N$ channel-independent generators are first trained. These modules generate single-channel time series $\widehat{T}_i$ for their respective channels, which are then concatenated to form $\widehat{\mathsf{T}}$. This concatenated time series $\widehat{\mathsf{T}}$, along with the real MTS $\mathsf{T}$, is then used to train the CCD and channel-independent generators.

# B  ADDITIONAL INFORMATION FOR SIMULATED DATASETS

**Simulated Datasets**  To effectively assess the performance and significance of CCD, we require a customizable time series dataset. Therefore, we utilized three synthetic datasets with dual channels as proposed by (Seyfi et al., 2022).

*Simple sine* The formula for a basic sine function is given as: $x = A\sin(2\pi f t) + \epsilon$, where $A$, $f$, and $\epsilon$ are hyperparameters. Across all time series data, channel 1 maintains a frequency $f$ of 0.01, while channel 2's frequency is set to 0.005. For the first data type, the amplitude $A$ is sampled from $N(0.4, 0.05)$, and for the second type, it is sampled from $N(0.6, 0.05)$.

*Sine with frequency changes* This variant is derived from a basic sine wave with a doubled frequency at the midpoint of the time series. By altering the frequency, we can observe how the model generates data with varying frequencies.

*Anomalies* Anomalies are generated from basic sine waves by replacing the original data at the midpoint of the time series with Gaussian noise. This allows us to evaluate the model's performance in generating data with outliers.

**Parameters**  The CCD module was configured with $k = 3$. Moreover, the training epochs were fixed at 500, simulating a resource and time-limited scenario, enabling a more effective evaluation of CCD.

**Metrics**  Following (Seyfi et al., 2022), we conducted a quantitative comparison of the correlation matrices between the two channels using several metrics: (1) Mean Squared Error (MSE), (2) Frobenius norm (FN), (3) Spearman's $\rho$, and (4) Kendall's $\tau$. The MSE and Frobenius norm metrics indicate a higher similarity between the correlation matrices of the real dataset and the generated synthetic dataset when smaller values are obtained. On the other hand, Spearman's coefficient and Kendall's coefficient approach a value of 1 as the similarity increases.

# C  DATASET SELECTION

**Dataset Selection.**  To ensure reproducibility and mitigate biases or oversimplification in our evaluations, we exclusively employ publicly available, real-world datasets. It is crucial to emphasize

Table 8: The statistics of the five datasets.

| Datasets | $R$ | $l$ | $N$ | Domain |
|---|---|---|---|---|
| Stock (Yoon et al., 2019) | 3,294 | 24 | 6 | Financial |
| Stock Long (Yoon et al., 2019) | 3,204 | 125 | 6 | Financial |
| Energy (Candanedo, 2017) | 17,739 | 24 | 28 | Appliances |
| Energy Long (Candanedo, 2017) | 17,649 | 125 | 28 | Appliances |
| EEG (Roesler, 2013) | 13,366 | 128 | 14 | Medical |
| DLG (Hutchins, 2006) | 246 | 14 | 20 | Traffic |
| Air (Zheng et al., 2015) | 7731 | 168 | 6 | Sensor |

that our objective is not to accumulate an exhaustive collection of datasets, but rather to curate a diverse set encompassing multiple domains, showcasing varied data statistics and distributions. Table 8 summarizes their statistics. Below, we provide a brief description of each dataset.

- **Stock (Yoon et al., 2019).** It comprises daily historical Google stock data from 2004 to 2019, including volume and high, low, opening, closing, and adjusted closing prices.
- **Stock Long (Yoon et al., 2019).** It is identical to the Stock dataset but with a sequence length of 125.
- **Energy (Candanedo, 2017).** It includes information on appliance's energy use in a low-energy building.
- **Energy Long (Candanedo, 2017)**. It is identical to the Energy dataset but with a sequence length of 125.
- **EEG (Roesler, 2013).** It is with the measurements derived from ElectroEncephaloGraphy (EEG) data captured by Emotiv EEG Neuroheadset. It helps to understand brainwave patterns, especially those under specific cognitive conditions or stimuli.
- **Dodgers Loop Game (DLG) Hutchins (2006).** It consists of loop sensor data from the Glendale on-ramp for the 101 North freeway in Los Angeles.
- **D9: Air Zheng et al. (2015).** It has air quality, meteorological, and weather forecast data from 4 major Chinese cities: Beijing, Tianjin, Guangzhou, and Shenzhen from 2014/05/01 to 2015/04/30.

This study selected five datasets from different real-world domains: the Stock dataset from the financial domain, the Energy dataset from the energy sector, the EEG dataset from biological signals, the DLG dataset from traffic, and the Air dataset. To better investigate the impact of subsequence length on the model, we set different values of l for the Stock and Energy datasets. Therefore, our method is applicable to real cases or practical applications.

## D    EVALUATION MEASURE SUITE

Numerous metrics are available to assess the quality of TSG (Time Series Generation) methods, which commonly adhere to principles such as diversity, fidelity, and utility.

**Model-based Measures**    These measures primarily follow the TSTR scheme (Esteban et al., 2017; Jordon et al., 2018), wherein synthetically generated series are utilized to train a post-hoc neural network, which is then evaluated on the original time series.

- **Discriminative Score (DS) (Yoon et al., 2019).** This study utilizes a post-hoc time-series classification model, leveraging 2-layer GRUs or LSTMs, to discern between original and generated series (Yoon et al., 2019). The original series are denoted as "real," whereas the generated series are labeled as "synthetic." Subsequently, an RNN classifier is trained using these labels. The fidelity of the generation model is assessed by measuring the classification error on a separate test set.
- **Predictive Score (PS) (Yoon et al., 2019).** It focuses on training a post-hoc time series prediction model using synthetic data (Yoon et al., 2019). The model leverages GRUs or LSTMs to predict either the temporal vectors of each input series for future steps (Yoon et al., 2019;

Jarrett et al., 2021) or the entire vector (Jeon et al., 2022). To assess performance, the model is evaluated on the original dataset using the mean absolute error metric.

**Feature-based Measures**   These measures aim to capture inter-series correlations and temporal dependencies, evaluating the extent to which the generated time series preserves the original characteristics. Feature-based measures offer a distinct advantage by providing clear and deterministic results, ensuring an unambiguous assessment of the quality of the generated time series.

- **Marginal Distribution Difference (MDD) (Ni et al., 2021).** This measure calculates empirical histograms for each dimension and time step in the generated series. The bin centers and widths from the original series are used for this purpose. By computing the average absolute difference between these histograms and those of the original series across bins, it assesses the alignment of the distributions between the original and generated series.
- **AutoCorrelation Difference (ACD) (Lai et al., 2018).** This measure calculates the autocorrelation of both the original and generated time series and determines their difference (Parzen, 1963; Lai et al., 2018). By comparing the autocorrelations, we can assess the preservation of dependencies in the generated time series.
- **Skewness Difference (SD).** In addition to ACF, this study incorporates statistical measures to assess the quality of the generated time series (Wang et al., 2023). One such measure is skewness, which quantifies the distribution asymmetry of a time series and is crucial for analyzing its marginal distribution. Given the mean (standard deviation) of the train time series $\boldsymbol{T}_s^{tr}$ as $\boldsymbol{\mu}_s^{tr}$ ($\boldsymbol{\sigma}_s^{tr}$) and the generated time series $\boldsymbol{T}_s^{gen}$ as $\boldsymbol{\mu}_s^{gen}$ ($\boldsymbol{\sigma}_s^{gen}$), we evaluate the fidelity of $\boldsymbol{T}_s^{gen}$ by computing the skewness difference between them as:

$$SD = \left| \frac{\mathbb{E}[(\boldsymbol{T}_s^{gen} - \boldsymbol{\mu}_s^{gen})^3]}{\boldsymbol{\sigma}_s^{gen3}} - \frac{\mathbb{E}[(\boldsymbol{T}_s^{tr} - \boldsymbol{\mu}_s^{tr})^3]}{\boldsymbol{\sigma}_s^{tr3}} \right|. \tag{8}$$

- **Kurtosis Difference (KD).** Similar to skewness, kurtosis is employed to evaluate the tail behavior of a distribution, uncovering extreme deviations from the mean. Using notations from Equation 8, the kurtosis difference between $\boldsymbol{T}_s^{tr}$ and $\boldsymbol{T}_s^{gen}$ is calculated as:

$$KD = \left| \frac{\mathbb{E}[(\boldsymbol{T}_s^{gen} - \boldsymbol{\mu}_s^{gen})^4]}{\boldsymbol{\sigma}_s^{gen4}} - \frac{\mathbb{E}[(\boldsymbol{T}_s^{tr} - \boldsymbol{\mu}_s^{tr})^4]}{\boldsymbol{\sigma}_s^{tr4}} \right|. \tag{9}$$

**Training Efficiency**   Training efficiency plays a critical role, especially in scenarios that require fast time series generation methods or when computational resources are limited. However, only a limited number of studies, such as (Desai et al., 2021; Jeon et al., 2022), have been utilized for evaluation in this particular context.

- **Training Time.** Training time, referring to the wall clock time required for training a time series generation (TSG) method, is a crucial metric for evaluating and deploying TSG methods. It holds significant importance due to economic considerations.

**Distance-based Measures**   To address the challenges associated with data synthesis (DS) and privacy preservation (PS), we propose the integration of two distance-based measures as a means of achieving an efficient and deterministic evaluation.

- **Euclidean Distance (ED).** For each original series $\boldsymbol{s}^{tr} = (x_1, \cdots, x_l)$ and its generated series $\boldsymbol{s}^{gen} = (y_1, \cdots, y_l)$, $ED = \sqrt{\sum_{i=1}^{l}(x_i - y_i)^2}$. The mean of the Euclidean distance (ED) is computed for all series and samples. As the input time series has been preprocessed to fall within the range of $[0, 1]$, ED allows for a deterministic evaluation of the similarity between $\boldsymbol{s}^{gen}$ and $\boldsymbol{s}^{tr}$. It facilitates a value-wise comparison of the time series.
- **Dynamic Time Warping (DTW) (Berndt & Clifford, 1994).** In order to account for alignment, we incorporate DTW to capture the optimal alignment between series, regardless of their pace or timing. The alignment facilitated by DTW provides valuable insights into the predictive quality of the generated series. Additionally, studies such as (Shokoohi-Yekta et al., 2017) have demonstrated that multi-dimensional DTW can enhance downstream classification tasks, making it a discriminative measure.

By leveraging the metrics of ED and DTW, we can efficiently and effectively assess the quality of generated time series. These metrics provide streamlined alternatives to evaluate time series generation, with similar goals as those of DS and PS.

## D.1 AED AND AWD

Time series generation tasks consider two key factors: Fidelity and Correlation Preservation. Fidelity aims to generate results that match the distribution of the original time series while avoiding mode collapse. We use the Average Wasserstein Distance (AWD) to reflect the diversity of the generated results. A smaller AWD indicates a closer resemblance to the true distribution and better fidelity. Correlation Preservation aims to maintain the same channel correlations as the original time series for different channels of the generated results. We map the generated time series onto a two-dimensional plane and calculate the Average Euclidean Distance (AED) between the generated series and the line with an amplitude and slope of 1. We use AED to measure the correlation preservation of the generated results.

## E BASELINES

We present a detailed analysis of five representative time series generation (TSG) methods that are based on three foundational generative models. To ensure experimental fairness, all experiments are conducted on a machine with Intel® Core® i9 12900K CPU @ 5.20 GHz, 64 GB memory, and NVIDIA GeForce RTX 3090.

**Pure GAN-based Methods** Early studies (Mogren, 2016; Esteban et al., 2017) incorporated vanilla GAN architectures originally designed for image generation and combined them with neural networks such as RNN and LSTM, specifically tailored for sequential data. Subsequent research has been dedicated to pioneering techniques that adapt to time series data and enhance performance.

- **RTSGAN (Pei et al., 2021).** RTSGAN integrates an autoencoder into GANs and focuses on generating time series with variable lengths while effectively handling missing data.
- **COSCI-GAN (Seyfi et al., 2022).** COSCI-GAN is specifically designed to explicitly capture the complex dynamical patterns within each series, with a focus on preserving the relationships among channels or features.

**Pure VAE-based Methods** VAE-based methods commonly leverage variational inference to effectively capture temporal features. These methods are known for their efficiency and potential interpretability.

- **TimeVAE (Desai et al., 2021).** TimeVAE extends the application of Variational Autoencoders (VAEs) to general-purpose time series generation. It incorporates convolutional operations and enhances interpretability through time series decomposition techniques.
- **TimeVQVAE (Lee et al., 2023).** TimeVAE incorporates the Short-Time Fourier Transform (STFT) to decompose input time series into low-frequency and high-frequency components. It further enhances the modeling of these components by integrating Vector Quantization with VAEs (Lee et al., 2023), ensuring the preservation of both the general shape and specific details of the time series.

**Mixed-Type Methods** Recent advancements in time series generation (TSG) have explored mixed-type methods, which involve combining flow-based models with techniques such as Discrete Fourier Transform (DFT) or Ordinary Differential Equations (ODEs). Additionally, these methods have been integrated with Generative Adversarial Networks (GANs) or Variational Autoencoders (VAEs) to further enhance their capabilities.

- **LS4 (Zhou et al., 2023).** LS4 is derived from deep state-space models and integrates stochastic latent variables to augment the model's capacity while leveraging the training objectives of Variational Autoencoders (VAEs).

# F    RESULTS ON SYNTHETIC DATASET

**Experimental Setup**    To effectively assess the performance and significance of CCD, we require a customizable time series dataset. Therefore, we utilized three synthetic datasets (*Simple sine*, *Sine with frequency changes*, and *Anomalies*) with dual channels as proposed by (Seyfi et al., 2022). Following (Seyfi et al., 2022), we conducted a quantitative comparison of the correlation matrices between the two channels using several metrics: (1) Frobenius norm (FN), (2) Spearman's $\rho$, and (3) Kendall's $\tau$. The detailed information on datasets, parameters, and metrics can be found in Appendix Section B.

**Comparison with COSCIGAN**    We compare our method with the state-of-the-art (SoTA) channel-independent method to demonstrate the performance improvements achieved by our method. To ensure experimental fairness, CCDGAN and COSCIGAN utilize the same single-channel generator and single-channel discriminator. The single-channel generator consists of a 1-layer LSTM network and three linear layers, while the single-channel discriminator consists of four linear layers. COSCIGAN's CD consists of four linear layers. Table 9 presents the results, highlighting significant advancements in three key metrics. For the Anomalies dataset, our performance in Kendall's $\tau$ falls short of COSCIGAN. This discrepancy can be attributed to Kendall's $\tau$'s focus on order correlation while remaining insensitive to outliers. Given the Anomalies dataset's substantial number of anomalies, the resulting weaker order correlation adversely impacts the fairness of metric evaluation.

Table 9: Results of CCDGAN and COSCIGAN on simulated datasets, where bold indicates methods that perform well in the respective metric. The performance of the central discriminator was evaluated by employing the same channel generator. $\uparrow$ represents the larger the value, the better, while $\downarrow$ represents the smaller the value, the better.

| Dataset | Method | FN$\downarrow$ | $\rho\uparrow$ | $\tau\uparrow$ |
|---------|--------|------|------|------|
| Simple Sine | CCDGAN | **3.725** | **0.667** | **0.021** |
| | COSCIGAN | 5.002 | 0.208 | -0.01 |
| Freq changes | CCDGAN | **1.868** | **0.659** | **-0.009** |
| | COSCIGAN | 2.285 | -0.075 | -0.05 |
| Anomalies | CCDGAN | **2.041** | **0.767** | -0.005 |
| | COSCIGAN | 2.531 | 0.032 | **0.02** |

Table 10:  The types of central discriminator (MLP or CCD) on the performance of channel independent framework. The smaller the values for AWD and AED, the better.

| Dataset | CD type | AWD$\downarrow$ | AED$\downarrow$ |
|---------|---------|------|------|
| Simple Sine | None | **0.047** | 0.133 |
| | MLP | 0.08 | 0.018 |
| | CCD | 0.055 | **0.014** |
| Freq changes | None | **0.04** | 0.077 |
| | MLP | 0.068 | 0.024 |
| | CCD | 0.061 | **0.017** |
| Anomalies | None | **0.054** | 0.077 |
| | MLP | 0.073 | 0.077 |
| | CCD | 0.066 | **0.071** |

Table 11: The results of different channel-independent generators with CCD. Among them, channel-independent GANs based on MLP, GRU, and TimeGAN were employed. We also utilized VAE to investigate whether non-GAN models, apart from GAN, can adopt the channel-independent approach.

| Dataset | Model | FN$\downarrow$ | $\rho\uparrow$ | $\tau\uparrow$ |
|---------|-------|------|------|------|
| Simple Sine | MLP | 6.737 | 0.122 | -0.108 |
| | GRU | 5.509 | 0.235 | -0.01 |
| | TimeGAN | 5.268 | 0.288 | -0.007 |
| | VAE | 7.338 | 0.170 | -0.016 |
| Freq changes | MLP | 3.004 | -0.144 | -0.198 |
| | GRU | 2.291 | -0.081 | -0.059 |
| | TimeGAN | 2.334 | -0.118 | -0.124 |
| | VAE | 5.004 | -0.104 | -0.037 |
| Anomalies | MLP | 4.889 | -0.1 | -0.016 |
| | GRU | 2.531 | 0.039 | 0.032 |
| | TimeGAN | 3.829 | 0.027 | 0.107 |
| | VAE | 3.482 | 0.002 | 0.021 |

**Analysis of Channel Independent Generator**    We explore the impact of different single-channel TSG modules on the overall framework. Considering the channel independence, we selected MLP, GRU, TimeGAN, and VAE as our channel-independent TSG modules incorporating the CCD as the central discriminator.

If we select pure VAE-based methods to be single-channel time series generation module, the objectives of these three components can be summarized as follows:

$$\min_{\theta_i} \max_{\alpha} \mathcal{L}(\text{VAE}_{i,\theta_i}, CCD_{\alpha}) = \mathcal{L}_{\text{VAE}_{i,\theta_i}} + \gamma \cdot \log(1 - CCD_{\alpha}(\text{VAE}_{i,\theta_i}(z), \text{VAE}_{j\neq i}(z)))) \quad (10)$$

$$\mathcal{L}_{\text{VAE}_{i,\theta_i}} = -\mathbb{E}_{i,\theta_i, q(z|t_i)}[\log p(t_i|z)] + \beta \cdot \text{KL}_{i,\theta_i}, (q(z|t_i)\|p(z)) + \gamma \cdot \log(CCD_{\alpha}(t_i)) \quad (11)$$

where $-\mathbb{E}_{q(z|t_i)}[\log p(t_i|z)]$ represents the reconstruction loss in the original loss function of VAE. Here, $q(z|t_i)$ represents the approximate posterior distribution of latent variables generated by the encoder, and $p(t_i|z)$ represents the reconstruction data distribution produced by the decoder. $\text{KL}(q(z|t_i)\|p(z))$ represents the KL divergence loss, where $p(z)$ represents the prior distribution, typically assumed to be a multivariate Gaussian distribution. $\beta$ and $\gamma$ are hyperparameters used to balance the KL divergence loss and the CCD loss.

Table 11 indicates that choosing alternative modules for the single-channel TSG modules is a viable approach. Regarding the module selection, it is advisable to opt for methods that exhibit superior performance in channel-mixing techniques while striking a balance between performance and computational resources. We also analyzed the performance of the entire framework when incorporating the non-GAN module VAE as the single-channel TSG module. Using VAE yields inferior results compared to GAN-based approaches. This discrepancy can be attributed to the possibility that the current training objectives of the channel-independent framework may not be suitable for VAE-based methods.

**Ablation Study of CCD**    We verified the effectiveness of the CCD module in capturing temporal dependencies within multivariate time series, both within and across periods. The Average Wasserstein Distance (AWD) was used to measure the diversity of generated results, while the Average Euclidean Distance (AED) assessed correlation preservation.

A channel-independent GAN method served as the baseline, with identical settings for all channel generators as described in Section F. The CD type used 'None' to denote the absence of a CD module and 'MLP' for the MLP-based central discriminator, which consists of consists of four linear layers. It concatenates the time series ($l \times N$) horizontally, resulting in a tensor with dimensions $l \times N$.

Table 10 shows that achieving both high fidelity and strong correlation preservation is challenging. While the central discriminator improves correlation, it may slightly reduce fidelity. In contrast, CCD outperforms the MLP-based central discriminator in both fidelity and correlation preservation.

# G    RESULTS ON REAL DATASET

The experimental results for Section 5.5 are included in the supplementary materials due to space limitations.

# H    CHANNEL INDEPENDENT TIMEVAE ADDING CCD

We utilized TimeVAE, a VAE-based model, and incorporated CCD loss into the loss function to investigate whether non-GAN models, apart from GAN, can adopt the channel-independent approach. In the following, bold indicates methods that perform well in the respective metric.

# I    ADDITIONAL INFORMATION FOR CCDGAN V.S. DIFFUSION-BASED METHODS

**Baselines**    We compared CCDGAN with three diffusion model-based approaches: Diffusion-TS(Yuan & Qiao, 2024), DiffWave(Kong et al., 2021), and DiffTime(Coletta et al., 2023). These methods, including Diffusion-TS, DiffWave, and DiffTime, are all based on DDPM, with modifications tailored to the characteristics of time series data.

Table 12: Detail results about Figure 3 on Stock dataset

| Methods Metrics | CCD | TimeVQVAE | TimeVAE | COSCI-GAN | LS4 | RTSGAN |
|---|---|---|---|---|---|---|
| MDD↓ | **0.315** | 1.186 | 0.327 | 0.334 | 0.502 | 0.319 |
| ACD↓ | **0.015** | 0.017 | 0.078 | 0.037 | 0.031 | 0.029 |
| SD↓ | **0.127** | 0.059 | 0.4 | 0.217 | 0.215 | 0.138 |
| KD↓ | **0.446** | 0.458 | 2.54 | 1.647 | 0.958 | 0.771 |
| ED↓ | 1.068 | **1.051** | 1.088 | 1.101 | 1.098 | 1.075 |
| DTW↓ | **2.772** | 2.791 | 2.787 | 3.014 | 2.781 | 2.784 |
| PS↓ | **0.073** | 0.094 | 0.084 | 0.086 | 0.109 | 0.088 |
| DS↓ | 0.327 | **0.163** | 0.177 | 0.396 | 0.482 | 0.339 |
| Time | 52min11s | 1h43min | **1min33s** | 55min35s | 47min50s | 1h03min |

Table 13: Detail results about Figure 3 on Stock Long dataset

| Methods Metrics | CCD | TimeVQVAE | TimeVAE | COSCI-GAN | LS4 | RTSGAN |
|---|---|---|---|---|---|---|
| MDD↓ | **0.451** | 1.207 | 0.455 | 0.457 | 0.482 | 0.516 |
| ACD↓ | **0.128** | 0.143 | 0.235 | 0.131 | 0.135 | 0.157 |
| SD↓ | **0.206** | 0.303 | 0.528 | 0.215 | 0.069 | 0.238 |
| KD↓ | 2.223 | 0.782 | 2.541 | 1.414 | **0.569** | 1.115 |
| ED↓ | **2.388** | 2.697 | 2.486 | 2.539 | 2.722 | 2.532 |
| DTW↓ | **6.229** | 6.269 | 6.319 | 6.625 | 6.698 | 6.626 |
| PS↓ | 0.093 | 0.112 | 0.086 | 0.095 | 0.093 | **0.084** |
| DS↓ | 0.314 | 0.244 | **0.124** | 0.399 | 0.471 | 0.382 |
| Time↓ | 49min | 4h41min | **56s** | 57min | 44min | 1h40min |

Table 14: Detail results about Figure 3 on Energy dataset

| Methods Metrics | CCD | TimeVQVAE | TimeVAE | COSCI-GAN | LS4 | RTSGAN |
|---|---|---|---|---|---|---|
| MDD↓ | **0.289** | 0.328 | 0.386 | 0.307 | 0.294 | 0.306 |
| ACD↓ | **0.023** | 0.074 | 0.118 | 0.028 | 0.075 | 0.099 |
| SD↓ | **0.062** | 0.129 | 0.138 | 0.07 | 0.218 | 0.34 |
| KD↓ | 0.572 | **0.421** | 0.674 | 0.588 | 0.824 | 0.723 |
| ED↓ | 0.966 | 0.956 | 0.987 | 0.972 | 0.92 | **0.775** |
| DTW↓ | 6.045 | 6.025 | 5.799 | 6.267 | 5.928 | **4.984** |
| PS↓ | **0.247** | 0.252 | 0.288 | 0.256 | 0.492 | 0.311 |
| DS↓ | 0.419 | **0.335** | 0.487 | 0.469 | 0.471 | 0.488 |
| Time↓ | 2h32min | 3h27min | **15min19s** | 2h11min | 1h38min | 1h9min |

Table 15: Detail results about Figure 3 on Energy Long dataset

| Methods Metrics | CCD | TimeVQVAE | TimeVAE | COSCI-GAN | LS4 | RTSGAN |
|---|---|---|---|---|---|---|
| MDD↓ | **0.319** | 0.413 | 0.431 | 0.328 | 0.783 | 0.433 |
| ACD↓ | **0.103** | 0.123 | 0.27 | 0.109 | 0.29 | 0.403 |
| SD↓ | **0.088** | 0.153 | 0.183 | 0.093 | 0.592 | 0.192 |
| KD↓ | 0.289 | **0.185** | 0.621 | 0.489 | 6.735 | 0.823 |
| ED↓ | **2.017** | 2.189 | 2.035 | 2.297 | 2.342 | 2.349 |
| DTW↓ | 13.377 | 13.549 | **12.527** | 13.776 | 13.788 | 14.325 |
| PS↓ | 0.311 | 0.253 | 0.289 | 0.254 | 0.488 | **0.253** |
| DS↓ | **0.477** | 0.492 | 0.499 | 0.483 | 0.486 | 0.496 |
| Time↓ | 1h57min | 2h58min | **43min** | 1h42min | 3h05min | 2h24min |

Table 16: Detail results about Figure 3 on EEG dataset

| Methods
Metrics | CCD | TimeVQVAE | TimeVAE | COSCI-GAN | LS4 | RTSGAN |
|---|---|---|---|---|---|---|
| MDD↓ | **0.208** | 0.214 | 0.347 | 0.316 | 0.249 | 0.233 |
| ACD↓ | **0.043** | 0.068 | 0.148 | 0.047 | 0.143 | 0.076 |
| SD↓ | **0.167** | 0.188 | 0.241 | 0.174 | 0.381 | 0.27 |
| KD↓ | 0.509 | **0.361** | 0.958 | 0.662 | 1.51 | 0.663 |
| ED↓ | **1.331** | 1.693 | 1.517 | 1.831 | 1.578 | 1.762 |
| DTW↓ | 6.263 | 6.371 | 6.538 | 7.182 | **6.121** | 7.049 |
| PS↓ | 0.041 | **0.033** | 0.039 | 0.045 | 0.218 | 0.041 |
| DS↓ | 0.442 | **0.339** | 0.475 | 0.451 | 0.491 | 0.404 |
| Time↓ | 2h02min | 4h30min | **42min** | 2h12min | 3h51min | 2h12min |

Table 17: Channel Independent TimeVAE adding CCD.

| Dataset | Model | MSE↓ | FN↓ | $\rho\uparrow$ | $\tau\uparrow$ |
|---|---|---|---|---|---|
| Simple Sine | Without CCD | 0.167 | 9.071 | -0.019 | -0.129 |
| | With CCD | **0.121** | **7.338** | **0.17** | **-0.016** |
| Freq changes | Without CCD | 0.179 | 6.382 | -0.191 | -0.235 |
| | With CCD | **0.123** | **5.004** | **-0.104** | **-0.037** |
| Anomalies | Without CCD | 0,236 | 7.711 | -0.136 | -0.113 |
| | With CCD | **0.143** | **3.482** | **0.002** | **0.021** |

Table 18: The results of different methods on 10% Energy dataset.

| Methods | CCD | TimeVQVAE | TimeVAE | COSCI-GAN | LS4 | RTSGAN |
|---|---|---|---|---|---|---|
| MDD↓ | 0.312 | 0.471 | 0.403 | 0.529 | 0.31 | 0.477 |
| ACD↓ | 0.19 | 0.173 | 0.192 | 0.232 | 0.286 | 0.308 |
| SD↓ | 0.381 | 0.272 | 0.386 | 0.394 | 0.573 | 0.407 |
| KD↓ | 0.697 | 0.494 | 0.958 | 0.723 | 1.801 | 0.931 |
| ED↓ | 1.66 | 1.832 | 1.775 | 1.994 | 1.989 | 1.982 |
| DTW↓ | 6.833 | 6.92 | 6.986 | 7.636 | 7.146 | 7.534 |

Table 19: The results of different methods on 10% EEG dataset.

| Methods | CCD | TimeVQVAE | TimeVAE | COSCI-GAN | LS4 | RTSGAN |
|---|---|---|---|---|---|---|
| MDD↓ | 0.441 | 0.404 | 0.432 | 0.783 | 0.528 | 0.476 |
| ACD↓ | 0.13 | 0.092 | 0.081 | 0.125 | 0.199 | 0.152 |
| SD↓ | 0.301 | 0.371 | 0.366 | 0.473 | 0.542 | 0.492 |
| KD↓ | 0.797 | 0.52 | 1.025 | 0.992 | 1.611 | 0.835 |
| ED↓ | 1.405 | 1.933 | 1.671 | 1.915 | 1.883 | 1.924 |
| DTW↓ | 6.835 | 6.711 | 6.709 | 7.529 | 6.802 | 7.635 |

**Parameters**    For Diffusion-TS, DiffWave and DiffTime, to ensure the fairness of the experiments, we strived to maintain consistency in parameter settings. We chose 4 attention heads, each with a dimension of 16, and selected 2 encoder and decoder layers.

**Datasets**    For dataset selection, we chose the previously mentioned Stock and Energy datasets. For the Sine dataset, we opted for the sine wave dataset provided in (Yoon et al., 2019), which is channel-independent and exhibits more diverse variations.

**Evaluation Metrics**    1) **Context-Fréchet Inception Distance (Context-FID) score** (Paul et al., 2022) quantifies the quality of the synthetic time series samples by computing the difference between representations of time series that fit into the local context; 2) **Correlational score** (CS) (Ni et al., 2020) uses the absolute error between cross correlation matrices by real data and synthetic data to assess the temporal dependency.

2) **Correlational Score** To mitigate the challenges associated with DS and PS, we propose the incorporation of distance-based measures to provide an efficient, deterministic evaluation.

3) **Discriminative Score** For a quantitative measure of similarity, we train a post-hoc time-series classification model (by optimizing a 2-layer LSTM) to distinguish between sequences from the original and generated datasets. First, each original sequence is labeled real, and each generated sequence is labeled not real. Then, an off-the-shelf (RNN) classifier is trained to distinguish between the two classes as a standard supervised task.

4) **Predictive Score** In order to be useful, the sampled data should inherit the predictive characteristics of the original. In particular, we expect TimeGAN to excel in capturing conditional distributions over time. Therefore, using the synthetic dataset, we train a post-hoc sequence-prediction model (by optimizing a 2-layer LSTM) to predict next-step temporal vectors over each input sequence. Then, we evaluate the trained model on the original dataset.

5) **Training Time** It refers to the wall clock time for training a TSG method. It is a vital measure for evaluating and deploying TSG methods due to economic considerations.

6) **2000 Data Sampling Time** It evaluates the time spent by the model in generating data, which was not previously emphasized, but significantly affects the user experience with the model. Additionally, methods based on diffusion models require longer generation times, leading to increased attention to sampling time.

## J    ANALYSIS OF CHANNEL-INDEPENDENT FRAMEWORK

Table 20: Ablation study of Period and Conv2D modules on the Stock and Energy datasets.

| Dataset | Stock | | | Energy | | |
|---|---|---|---|---|---|---|
| Metrics | No Period | No Conv2d | CCD | No Period | No Conv2d | CCD |
| MDD↓ | 0.433 | 0.431 | **0.305** | 0.226 | 0.395 | **0.219** |
| ACD↓ | 0.022 | 0.039 | **0.015** | 0.037 | 0.044 | **0.023** |
| SD↓ | 0.149 | 0.142 | **0.127** | 0.29 | 0.285 | **0.272** |
| KD↓ | 0.829 | 0.848 | **0.556** | 0.781 | 0.801 | **0.748** |
| ED↓ | 1.103 | 1.097 | **1.068** | 0.994 | 0.979 | **0.966** |
| DTW↓ | 4.442 | 4.39 | **4.252** | 7.131 | 7.195 | **7.045** |
| PS↓ | 0.09 | 0.127 | **0.073** | 0.305 | 0.28 | **0.247** |
| DS↓ | 0.408 | 0.483 | **0.327** | 0.424 | 0.433 | **0.419** |

**Exploration of Different Channel Independent Generators**    Following (Seyfi et al., 2022), we conducted a quantitative comparison on three synthetic datasets (*Simple sine*, *Sine with frequency changes*, and *Anomalies*). We explore the impact of different channel independent generators on the overall framework. The detailed results can be found in Appendix Section F. TimeGAN is a better team for CCD than MLP, GRU, and VAE.

## J.1 ANALYSIS OF CCD

**Ablation Study of CCD**   We validated the effectiveness of the Period and Conv2D modules on the Stock(Yoon et al., 2019) and Energy(Candanedo, 2017) datasets. **No Period** indicates the removal of the Period module from CCD, avoiding manipulation of real and fake time series dimensions. **No Conv2D** denotes the replacement of the Conv2D block with three linear layers.

The results in Table 20 show that removing either the Period or Conv2D block decreases performance. This underscores the importance of both components in CCD, highlighting their role in enhancing and capturing various types of local information across different datasets.

**Exploration of Period Blocks in CCD**   We investigated the impact of using FFT versus Wavelet Transform in the Period Block. While FFT focuses solely on the frequency domain, the Wavelet Transform offers both frequency and time localization. Experiments on the Stock dataset, with consistent parameters except for frequency extraction, revealed minimal performance differences between FFT and Wavelet Transform, as shown in Table 21.

Table 21: Analysis of Period Block on Stock dataset. WT stands for Wavelet transform.

| Period Block | FFT | WT |
|---|---|---|
| MDD↓ | 0.315 | 0.313 |
| ACD↓ | 0.015 | 0.019 |
| SD↓ | 0.127 | 0.132 |
| KD↓ | 0.446 | 0.448 |
| ED↓ | 1.068 | 1.062 |
| DTW↓ | 2.772 | 2.779 |
| PS↓ | 0.073 | 0.086 |
| DS↓ | 0.327 | 0.319 |

Table 22: Analysis of Kernel size on Energy and Stock dataset.

| Dataset | Stock | | | | Energy | | | |
|---|---|---|---|---|---|---|---|---|
| Kernel Size | 1 | 3 | 5 | varied | 1 | 3 | 5 | varied |
| MDD↓ | 0.485 | 0.433 | 0.572 | **0.305** | 0.306 | 0.289 | 0.251 | **0.219** |
| ACD↓ | 0.171 | 0.211 | 0.193 | **0.015** | 0.049 | 0.051 | 0.052 | **0.023** |
| SD↓ | 0.174 | 0.16 | 0.152 | **0.127** | 0.296 | **0.267** | 0.284 | 0.272 |
| KD↓ | 0.807 | 0.719 | 0.692 | **0.556** | 0.912 | 0.959 | 0.761 | **0.700** |
| ED↓ | 1.102 | 1.079 | 1.092 | **1.068** | 0.992 | 0.979 | 0.971 | **0.966** |
| DTW↓ | 5.104 | 4.829 | 4.701 | **4.252** | 7.281 | 7.163 | 7.082 | **7.045** |
| PS↓ | 0.166 | 0.092 | 0.096 | **0.073** | 0.466 | 0.301 | 0.288 | **0.247** |
| DS↓ | 0.468 | 0.391 | 0.383 | **0.327** | 0.484 | 0.466 | 0.451 | **0.419** |

**Analysis of Kernel Size**   We investigated the selection of Conv2D kernel sizes, testing two strategies: fixed kernel sizes (1, 3, and 5) and varied kernel sizes defined by $kernel\ size = 2 \times i + 1$ for $i = 0, 1, \cdots, k$ with $k = 3$ as the number of Conv2D blocks. Results on the Stock and Energy datasets, shown in Table 22, demonstrate that the varied kernel size strategy outperforms the fixed kernel size approach. Stacking Conv2D modules with increasing kernel sizes allows the model to capture more diverse and richer feature representations.

The benefits of adopting such a strategy are as follows: 1) Compared to a fixed convolutional kernel, by stacking conv2d modules with increasing kernel sizes, the model can extract more diverse and richer feature representations. Different kernel sizes can capture different types of features. 2) By using smaller kernel sizes in the initial convolutions, the model can simultaneously focus on finer local features. As the conv2d blocks are stacked, the model needs to acquire broader global features.

**Effect of no. of Channels on CCD**   We explored the impact of the number of channels $N$, using the Energy dataset as an example. We randomly selected channel data with numbers 4, 8, and 12 in the Energy dataset. The results, as shown in Table 23, indicate that the number of channels has almost no impact on CCDGAN.

We investigated the impact of subsequence length $l$. The Stock dataset was divided into Stock ($l = 24$) and Stock Long ($l = 125$) lengths, and the Energy dataset was divided into Energy ($l = 24$) and Energy Long ($l = 125$). Experimental results are shown in Tables 9-12. As the subsequence length $l$ increases, the performance metrics of CCDGAN degrade but still outperform other methods.

Table 23: Analysis of number of channels on Energy dataset.

| No. of channels | 4 | 8 | 12 |
|---|---|---|---|
| MDD↓ | 0.314 | 0.315 | 0.315 |
| ACD↓ | 0.017 | 0.015 | 0.013 |
| SD↓ | 0.128 | 0.127 | 0.127 |
| KD↓ | 0.442 | 0.446 | 0.445 |
| ED↓ | 1.068 | 1.068 | 1.07 |
| DTW↓ | 2.773 | 2.772 | 2.773 |
| PS↓ | 0.071 | 0.073 | 0.081 |
| DS↓ | 0.32 | 0.327 | 0.331 |

Table 24: Analysis of number of layers on Stock dataset.

| No. of layers | 2 | 3 | 4 |
|---|---|---|---|
| MDD↓ | 0.307 | 0.315 | 0.318 |
| ACD↓ | 0.017 | 0.015 | 0.014 |
| SD↓ | 0.133 | 0.127 | 0.127 |
| KD↓ | 0.442 | 0.446 | 0.445 |
| ED↓ | 1.066 | 1.068 | 1.067 |
| DTW↓ | 2.775 | 2.772 | 2.78 |
| PS↓ | 0.08 | 0.073 | 0.079 |
| DS↓ | 0.331 | 0.327 | 0.325 |

**Effect of No. of convolution layers on CCD** For the number of convolution layers, we experimented with 2, 3, and 4 layers on the Stock dataset, while keeping other experimental settings and parameters constant. The results are shown in Table 24. The results indicate that setting the convolution layers to 3 yields relatively good experimental results on the Stock dataset. Therefore, this setting can be initially applied to other datasets, with adjustments made based on the actual situation.

## K TSGGUIDE VERSUS TSGBENCH

In contrast to TSGBench (Ang et al., 2023a), the updated sections are marked in blue.

### K.1 TSGBENCH

1. As a foundational step, we advocate for users to commence with VAE-based methods (e.g., TimeVAE and LS4). Their consistent leading performance and superior computational efficiency make them go-to choices for initial exploration.

2. In applications that emphasize autocorrelation or forecasting, such as predictive maintenance or stock market analysis, the ACD measure becomes crucial. Fourier Flow, which is recognized for maintaining temporal dependencies, is highly suitable for these scenarios. On the other hand, for capturing complex multi-variate relationships in datasets, COSCI-GAN is the recommended choice.

3. Subsequent considerations focus on dataset size and domain specificity. For small-sized datasets, RTSGAN and LS4, which excel in single DA, are strong choices. For heterogeneous datasets, or when the goal is to generate time series for a new target domain, TimeVAE and COSCI-GAN stand out for their effectiveness in cross DA.

4. Users can further fine-tune their method selection based on specific real-world application needs, which involves identifying the most relevant evaluation measures. In this case, Figure 1 serves as a valuable visual guide.

### K.2 TSGGUIDE

1. In Figure 3, VAE-based methods demonstrate faster training times and rank above average on several metrics. This makes VAE-based methods suitable for initial attempts due to their lower time requirements. Furthermore, the results in Tables 3, and Appendix Tables 18 and 19 show that VAE-based methods perform well on small datasets, making them applicable to various types of datasets. As a foundational step, we recommend that users start with the VAE-based method. Its consistently excellent computational efficiency makes it the preferred choice for initial exploration and handling of small data sets.

2. If we emphasize autocorrelation or forecasting, such as predictive maintenance or stock market analysis, the ACD measure becomes crucial. CCDGAN is highly suitable for these scenarios.

3. In Figure 3, CCD shows a leading performance on the majority of datasets, proving that CCDGAN performs well on real-world datasets, even though they come from different

domains. When the dataset originates from a novel domain or exhibits complex multivariate relationships, CCDGAN is a recommended choice. It shows excellent performance across data sets from various domains and outperforms other methods, particularly on datasets with a large number of variables.

4. In Table 5, diffusion model-based methods achieve certain advantages in metrics. However, these methods require significant training and sampling time, leading to higher computational costs. Therefore, if ample computational resources are available, diffusion model-based approaches should be considered. When computational resources and dataset are abundant and optimal results are desired, it is recommended to utilize Diffusion-TS. These methods typically exhibit stable training and yield superior performance.

5. Users can further adjust method selection based on specific application requirements, including identifying the appropriate channel-independent TSG module for CCDGAN (see Appendix Section F).

## L    PRE-TRAINING DATA DETAILS

We provide additional details regarding the pre-training data. In our approach, 4 is pre-trained on a large synthetic dataset of paired numeric and symbolic data, utilizing the data generation technique from Kamienny et al. (2022). Each example consists of a set of $N$ points $(\boldsymbol{x}, y) \in \mathbb{R}^{D+1}$ and an associated mathematical function $f(\cdot)$, such that $y = f(\boldsymbol{x})$. These examples are generated by first sampling a function $f$, followed by sampling $N$ numeric input points $\boldsymbol{x}_i; i = 1, \ldots, N \in \mathbb{R}^D$ from $f$, and then calculating the target value $y_i = f(\boldsymbol{x}_i)$.

### L.1    SAMPLING OF FUNCTIONS

To generate random functions $f$, we employ the strategy outlined in Kamienny et al. (2022), building random trees with mathematical operators as nodes and variables/constants as leaves. This process includes:

**Input Dimension Selection.** We begin by selecting the input dimension $D$ for the functions from a uniform distribution $\mathcal{U}(1, D_{max})$. This step ensures variability in the number of input variables.

**Binary Operator Quantity Selection.** Next, we determine the quantity of binary operators $b$ by sampling from $\mathcal{U}(D - 1, D + b_{max})$ and selecting $b$ operators randomly from the set $\mathcal{U}(+, -, \times)$. This step introduces variability in the complexity of the generated functions.

**Tree Construction.** Using the chosen operators and input variables, we construct binary trees, simulating the mathematical function's structure. The construction process is performed following the method proposed in Kamienny et al. (2022).

**Variable Assignment to Leaf Nodes.** Each leaf node in the binary tree corresponds to a variable, which is sampled from the set of available input variables ($x_d$ for $d = 1, \ldots, D$).

**Unary Operator Insertion.** Additionally, we introduce unary operators by selecting their quantity $u$ from $\mathcal{U}(0, u_{max})$ and randomly inserting them from a predefined set ($\mathcal{O}_u$) of unary operators where $\mathcal{O}_u = [\text{inv}, \text{abs}, \text{pow2}, \text{pow3}, \text{sqrt}, \text{sin}, \text{cos}, \text{tan}, \text{arctan}, \text{log}, \text{exp}]$.

**Affine Transformation.** To further diversify the functions, we apply random affine transformations to each variable ($x_d$) and unary operator ($u$). These transformations involve scaling ($a$) and shifting ($b$) by sampling values from $D_{\text{aff}}$. In other words, we replace $x_d$ with $ax_d + b$ and $u$ with $au + b$, where $(a, b)$ are samples from $D_{\text{aff}}$. This step enhances the variety of functions encountered during pre-training and ensures the model encounters a unique function each time, aiding in mitigating the risk of overfitting as well as memorization. We used ten functions to generate 100 datasets. The following are the functions we used: add, sub, mul, div, abs, inv, sqrt, log, exp, sin.

### L.2    SAMPLING OF DATAPOINTS

Once have generated a sample function $f$, we proceed to generate $N$ input points $x_i \in \mathbb{R}^D$ and calculate their corresponding target value $y_i = f(x_i)$. To maintain data quality and relevance, we follow the guidelines from Kamienny et al. (2022), which include: *Discarding and Restarting:* If any input point $x_i$ falls outside the function's defined domain or if the target value $y_i$ exceeds $10^{100}$, we discard the sample function and restart the generation process. This ensures that the model learns

meaningful and well-behaved functions. *Avoidance and Resampling:* Avoidance and resampling of out-of-distribution $x_i$ values provide additional insights into $f$ as it allows the model to learn its domain. This practice aids the model in handling input variations. *Diverse Input Distributions:* To expose the model to a broad spectrum of input data distributions, we draw input points from a mixture of distributions, such as uniform or Gaussian. These distributions are centered around $k$ randomly chosen centroids, introducing diversity and challenging the model's adaptability.

The generation of input points involves the following steps:

**Cluster and Weight Selection.** We start by sampling the number of clusters $k$ from a uniform distribution $\mathcal{U}(1, k_{max})$. Additionally, we sample $k$ weights $\{w_j \sim \mathcal{U}(0, 1)\}_{j=1}^{k}$, which are normalized to $\sum_j w_j = 1$.

**Cluster Parameters.** For each cluster, we sample a centroid $\mu_j \sim \mathcal{N}(0, 1)^D$, a vector of variances $\sigma_j \sim \mathcal{U}(0, 1)^D$, and a distribution shape $D_j$ from $\{\mathcal{N}, \mathcal{U}\}$ (Gaussian or uniform). These parameters define the characteristics of each cluster.

**Input Point Generation.** We sample $[w_j N]$ input points from the distribution $D_j(\mu_j, \sigma_j)$ for each cluster $j$. This sampling with different weights from different distributions ensures the sampling of a diverse set of input points with varying characteristics.

**Normalization.** Finally, all generated input points are concatenated and normalized by subtracting the mean and dividing by the standard deviation along each dimension.

