# OpenReview forum: "TSGGuide: Recommendation Guide for Multivariate Time Series Generation"
_ICLR.cc/2025/Conference — Submitted to ICLR 2025_

### Official Review · Reviewer_5rTZ · 2024-11-04

**Soundness:** 3
**Presentation:** 2
**Contribution:** 2
**Rating:** 5
**Confidence:** 3

**Summary:**

This paper addresses three issues in the existing MSTG recommendation guidance: inadequate evaluation, limited independent exploration of channels, and the inability of CCD to capture dependencies. The authors propose an improved solution for MSTG. The innovation lies in introducing an enhanced CCD that demonstrates strong practicality in small sample scenarios. Compared to previous recommendation guidance, the proposed enhanced CCD places greater emphasis on the independent channel framework and diffusion model methods, aiming to provide more refined recommendations for different contexts.

**Strengths:**

1. The paper is well-written, with clear and fluent language.
2. The experiments are comprehensive, and the details are thoroughly presented (exclude key hyperparameters though).
3. The overall methodology is easy to understand, and the enhanced CCD performs well.

**Weaknesses:**

1. Some equations are not clearly presented; for example, the notation in Eq. 4 is not standard.
2. Since hyperparameters are crucial for time series-related tasks, the lack of details on hyperparameters (including those for your method and the compared methods) makes replication difficult.
3. The paper feels somewhat lacking in originality. Notably, in Section 5, while three synthetic datasets are used, the motivation for using these datasets is insufficient.
4. In Section 5, the paper primarily explains the advantages of CCD on small sample data from the perspective of time series classification, but it lacks a discussion of other possible reasons. Moreover, it does not sufficiently explore the strengths and weaknesses of the diffusion model method and the CCD module in different scenarios.

**Questions:**

Plz see weakness.

---

> ### Author Response · Authors · 2024-11-25
>
> Thank you very much for your comments. Our peer-to-peer responses can be found below. Also, we have uploaded a corrected PDF version of the article.
>
> **answer to w1:**
> We have fixed these in the updated versions.
>
> **answer to w2:**
> Thank you for your suggestion. We have added the description of hyperparameters in Parameters in Section 3.1.
>
> **answer to w3:**
> We conducted experiments based on DLG and Air, which are from the traffic flow dataset and air dataset respectively. The experimental results are as follows:
>
> | Dataset | DLG |   |   |   |   |  |
> | --- | --- | --- | --- | --- | --- | --- |
> | Method | CCDGAN | TimeVQVAE | TimeVAE | COSCI-GAN | LS4 | RTSGAN |
> |  MDD | 0.284 | 0.293 | 0.301 | 0.241 | 0.227 | 0.238 |
> |  ACD | 0.121 | 0.162 | 0.164 | 0.137 | 0.117 | 0.178 |
> |  SD | 0.214 | 0.235 | 0.209 | 0.257 | 0.216 | 0.227 |
> |   KD | 12.341 | 12.25 | 12.175 | 12.275 | 12.377 | 12.375 |
> |  ED | 1.12 | 1.363 | 1.315 | 1.641 | 1.363 | 1.177 |
> | DTW | 2.346 | 2.367 | 2.476 | 2.316 | 2.375 | 2.438 |
> | PS | 0.415 | 0.488 | 0.446 | 0.461 | 0.425 | 0.427 |
> |  DS | 0.245 | 0.221 | 0.237 | 0.257 | 0.227 | 0.288|
>
> | Dataset | Air|   |   |   |   |  |
> | --- | --- | --- | --- | --- | --- | --- |
> | Method | CCDGAN | TimeVQVAE | TimeVAE | COSCI-GAN | LS4 | RTSGAN |
> |  MDD | 0.139 | 0.127 | 0.121 | 0.142 | 0.148 | 0.117 |
> | ACD | 0.109 | 0.117 | 0.109 | 0.117 | 0.115 | 0.126 |
> | SD | 0.356 | 0.328 | 0.387 | 0.382 | 0.361 | 0.371 |
> | KD | 8.147 | 8.278 | 8.178 | 8.187 | 8.169 | 8.171 |
> | ED | 0.816 | 0.828 | 0.827 | 0.826 | 0.829 | 0.877 |
> | DTW | 2.044 | 2.28 | 2.091 | 2.081 | 2.062 | 2.027 |
> | PS | 0.404 | 0.483 | 0.426 | 0.433 | 0.429 | 0.416 |
> | DS | 0.107 | 0.186 | 0.131 | 0.124 | 0.139 | 0.121|
>
> **Simulated data**. At the same time, in order to demonstrate the effectiveness of our method, we used the following scheme to generate a simulated time series dataset: we adopt a framework based on the principle Y=f(X). Specifically, the process begins with sampling sequences X from a predefined random mixture distribution.
>
> Next, we employ a binary tree structure to randomly generate mathematical expressions that define the function f(X). This mechanism ensures variability and diversity in the functional relationships represented by f.
>
> Finally, by forward-propagating the sampled sequences X through the generated mathematical expressions f(X), we obtain the corresponding output sequences Y=f(X).
> This methodology provides a flexible and systematic way to simulate complex time-series data with customizable properties, which is critical for evaluating the robustness and generalization capabilities of our proposed models. The details of this dataset can be found in Appendix L.
>
> We generated 100 simulated data sets and conducted experiments according to the experimental design in Figure 3 in the main text. The average results of the experiments are shown below:
>
> | Method| CCDGAN| TimeVQVAE| COSCI-GAN| LS4| Diffusion-TS |
> | -| -| -| -| -| - |
> | MDD$\downarrow$| 0.251| 0.282| 0.272| 0.258| **0.247** |
> | ACD$\downarrow$| 0.116| 0.128| 0.107| **0.105**| 0.114 |
> |  SD$\downarrow$| **0.241**| 0.259| 0.244| 0.248| 0.246 |
> | KD$\downarrow$| **11.01**| 11.017| 11.028| 11.027| 11.014  |
> | ED$\downarrow$| 1.153| 1.151| 1.158| 1.155| **1.151** |
> | DTW$\downarrow$| **2.466**| 2.469| 2.474| 2.471| 2.472 |
> | PS$\downarrow$| 0.164| **0.162**| 0.169| 0.175| 0.166 |
> | DS$\downarrow$| 0.205| 0.217| 0.212| 0.219| **0.202**|
>
> The results demonstrate that both the diffusion-based channel-mixing framework Diffusion-TS and the GAN-based channel-independent framework CCDGAN excel across multiple evaluation metrics, with each achieving top rankings in three categories. CCDGAN outperforms in two distance-based metrics, likely due to GANs' strength in capturing temporal distance characteristics. Meanwhile, Diffusion-TS exhibits balanced performance across all metrics, highlighting the overall effectiveness of diffusion-based generation models.
>
> **answer to w4:**
> This paper introduces a novel Central Discriminator (CCD) designed from the perspective of small-sample time series classification, which significantly outperforms COSCI-GAN (a channel-independent framework) and enhances the recommended priority of channel-independent frameworks. However, this finding is only one aspect of the recommendations provided in this paper and is not the core contribution. We have thoroughly discussed the strengths and limitations of CCD-GAN and diffusion models across different domains and scenarios and provided corresponding recommendation guidelines.

---

### Official Review · Reviewer_vfbR · 2024-11-04

**Soundness:** 2
**Presentation:** 1
**Contribution:** 2
**Rating:** 6
**Confidence:** 3

**Summary:**

This paper aims at providing guidelines to researchers on how to select different Multivariate Time Series Generation models and metrics based on the scenatios that researchers are working on. More specifically, the authors provide their empirical findings across dataset characteristics, computational resources, and metrics. Based on their own findings and motivations on analyzing the perofrmance on small datasets with GAN-based model, the authors further developed CCD, a light-scale architecture for small datasets. It successfully improve the performance of GAN-based model on this task and make them close, even comparable to VAE-based models, which is dominant on this task. Moreover, the efficiency of CCD-based GAN is better than diffusion based model.

**Strengths:**

This paper's content covers a pretty wide range. Its empirical findings include:

1. Understanding how we should select our models based on our dataset characteristics and computational resources

2. Improving the performance of GAN-based models on this task

The recommendations are useful for researchers to apply in their research

**Weaknesses:**

1. The experiments, especially compared to the main benchmark paper TSGBench, is not comprehensive enough. Many datasets included in TSGBench are not covered. The TSGBench uses 10 datasets, while this paper only make use of the Stock and Energy dataset. The generalizability of the recommendation on other domains is questionable. Some baselines are also missing.

2. The writing is so confusing. I seldom comment a paper with this issue. But this paper's contribution is quite unclear. I spent several days trying to understand the central contribution of this paper. In abstract, the authors said that they would like to provide a recommendation. I understand it as a kind of more instructive benchmark. But then they start to present CCD, which is a margnially novel improvement on the GAN-based model (a new kind of discriminator in GAN). After that they again start to do analysis. My understanding is that the authors propose a relatively new improvement on GAN-based models. But then they found that the proposed model is still defeated by VAE-based model on small-scale dataset and diffusion models when resource is sufficient. So they try to sell their work by embedding it into a so-called recommendation.

3. Some results in this recommendation is well-known. The finding 4 about diffusion model's large training and inference load is well-known.

**Questions:**

1. What is the main contribution of this paper? Do you want to sell a new method or provide a recommendation?

2. What are the performances of the CCD on other domains such as air quality?

3. About the training time, I hope to know more detailed analysis. For example, if method A is with longer training time of method B, is it because A need more iterations, or because its computational load on one sample is higher?

---

> ### Author Response · Authors · 2024-11-25
>
> Thank you very much for your comments. Our peer-to-peer responses can be found below. Also, we have uploaded a corrected PDF version of the article.
>
> **answer to w1:**
> We conducted experiments based on DLG and Air, which are from the traffic flow dataset and air dataset respectively. The experimental results are as follows:
>
> | Dataset | DLG |   |   |   |   |  |
> | --- | --- | --- | --- | --- | --- | --- |
> | Method | CCDGAN | TimeVQVAE | TimeVAE | COSCI-GAN | LS4 | RTSGAN |
> |  MDD | 0.284 | 0.293 | 0.301 | 0.241 | 0.227 | 0.238 |
> |  ACD | 0.121 | 0.162 | 0.164 | 0.137 | 0.117 | 0.178 |
> |  SD | 0.214 | 0.235 | 0.209 | 0.257 | 0.216 | 0.227 |
> |   KD | 12.341 | 12.25 | 12.175 | 12.275 | 12.377 | 12.375 |
> |  ED | 1.12 | 1.363 | 1.315 | 1.641 | 1.363 | 1.177 |
> | DTW | 2.346 | 2.367 | 2.476 | 2.316 | 2.375 | 2.438 |
> | PS | 0.415 | 0.488 | 0.446 | 0.461 | 0.425 | 0.427 |
> |  DS | 0.245 | 0.221 | 0.237 | 0.257 | 0.227 | 0.288|
>
> | Dataset | Air|   |   |   |   |  |
> | --- | --- | --- | --- | --- | --- | --- |
> | Method | CCDGAN | TimeVQVAE | TimeVAE | COSCI-GAN | LS4 | RTSGAN |
> |  MDD | 0.139 | 0.127 | 0.121 | 0.142 | 0.148 | 0.117 |
> | ACD | 0.109 | 0.117 | 0.109 | 0.117 | 0.115 | 0.126 |
> | SD | 0.356 | 0.328 | 0.387 | 0.382 | 0.361 | 0.371 |
> | KD | 8.147 | 8.278 | 8.178 | 8.187 | 8.169 | 8.171 |
> | ED | 0.816 | 0.828 | 0.827 | 0.826 | 0.829 | 0.877 |
> | DTW | 2.044 | 2.28 | 2.091 | 2.081 | 2.062 | 2.027 |
> | PS | 0.404 | 0.483 | 0.426 | 0.433 | 0.429 | 0.416 |
> | DS | 0.107 | 0.186 | 0.131 | 0.124 | 0.139 | 0.121|
>
> **Simulated data**. At the same time, in order to demonstrate the effectiveness of our method, we used the following scheme to generate a simulated time series dataset: we adopt a framework based on the principle Y=f(X). Specifically, the process begins with sampling sequences X from a predefined random mixture distribution.
>
> Next, we employ a binary tree structure to randomly generate mathematical expressions that define the function f(X). This mechanism ensures variability and diversity in the functional relationships represented by f.
>
> Finally, by forward-propagating the sampled sequences X through the generated mathematical expressions f(X), we obtain the corresponding output sequences Y=f(X).
> This methodology provides a flexible and systematic way to simulate complex time-series data with customizable properties, which is critical for evaluating the robustness and generalization capabilities of our proposed models. The details of this dataset can be found in Appendix L.
>
> We generated 100 simulated data sets and conducted experiments according to the experimental design in Figure 3 in the main text. The average results of the experiments are shown below:
>
> | Method| CCDGAN| TimeVQVAE| COSCI-GAN| LS4| Diffusion-TS |
> | -| -| -| -| -| - |
> | MDD$\downarrow$| 0.251| 0.282| 0.272| 0.258| **0.247** |
> | ACD$\downarrow$| 0.116| 0.128| 0.107| **0.105**| 0.114 |
> |  SD$\downarrow$| **0.241**| 0.259| 0.244| 0.248| 0.246 |
> | KD$\downarrow$| **11.01**| 11.017| 11.028| 11.027| 11.014  |
> | ED$\downarrow$| 1.153| 1.151| 1.158| 1.155| **1.151** |
> | DTW$\downarrow$| **2.466**| 2.469| 2.474| 2.471| 2.472 |
> | PS$\downarrow$| 0.164| **0.162**| 0.169| 0.175| 0.166 |
> | DS$\downarrow$| 0.205| 0.217| 0.212| 0.219| **0.202**|
>
> The results demonstrate that both the diffusion-based channel-mixing framework Diffusion-TS and the GAN-based channel-independent framework CCDGAN excel across multiple evaluation metrics, with each achieving top rankings in three categories. CCDGAN outperforms in two distance-based metrics, likely due to GANs' strength in capturing temporal distance characteristics. Meanwhile, Diffusion-TS exhibits balanced performance across all metrics, highlighting the overall effectiveness of diffusion-based generation models.
>
> We selected some representative data sets, which cover some commonly used fields and also cover data sets of different lengths.
>
> **answer to w2:**
> Essentially, ours is a research article that provides users with a sound recommendation guide. The proposed CCD is a combination of current techniques to overcome the defect that the central discriminator in the channel-independent generation method cannot handle the small sample time series classification problem. We then conclude that the recommendation priority of the channel-independent method should be improved. We believe that the recommendation guide is valuable to the field and users.
>
> **answer to w3:**
> For users who do not understand this field, these results and suggestions are very important, even if it is an obvious result.
>
> **answer to q1:**
> see answer to w2.
>
> **answer to q2:**
> see answer to w1.
>
> **answer to q3:**
> For the same type of methods, we used the same number of iterations, and our parameters are described in detail in Section 3.1, but the models do have differences in the number of parameters.

---

### Official Review · Reviewer_QRyW · 2024-11-04

**Soundness:** 1
**Presentation:** 2
**Contribution:** 1
**Rating:** 3
**Confidence:** 3

**Summary:**

The paper provides a recommendation framework for selecting suitable multivariate time series generation (MTSG) methods across various scenarios. It addresses existing gaps in previous frameworks, particularly by incorporating recent diffusion-based methods and enhancing the channel-independent framework using a central discriminator (CCD). The paper conducts extensive experiments on several datasets, including Stock, Energy, and EEG data, which varifies the effectiveness of the proposed model.

**Strengths:**

S1. The paper is well-organized, and each module is clearly explained.

S2. The CCD introduces frequency-domain transformations and a central discriminator to capture inter-channel dependencies.

**Weaknesses:**

W1. The technical contribution need to be clarified. The proposed method reshapes time series data and applies a 2D convolutional framework, followed by optimization with a basic GAN architecture. However, this approach lacks novelty and may fall short in capturing the complex inter-channel correlations inherent to multivariate time series. Reshaping the data and applying 2D convolutions does not seem to address the unique and complicated dependencies between channels. A more comprehensive module that learns correlations directly to capture correlations and periodicity might be more effective and adaptive. Moreover, The CCD relies on 2D convolutions, a standard technique in time series modeling (such as TCN). The paper does not sufficiently justify what specific innovation CCD brings beyond typical 2D convolutional methods.

W2. The experiments focus on stock, energy, and EEG datasets, which may not sufficiently represent the diversity of real-world MTSG applications. The results would be more convincing if the proposed approach were evaluated on datasets covering additional scenarios of spatial-temporal forecasting (such as OD flow and traffic forecasting), which would better validate the method’s generalizability and effectiveness.

W3. The baselines used for comparison are not up-to-date, with the latest being from 2023. More recent methods like TimeGAE [1] and DLT [2] should be included to conduct sufficient experiment. What’s more, the paper lacks a detailed hyperparameter experiments, such as the weighting factor gamma in Equation 6.

W4. The authors do not specify how the datasets are split into training, validation, and test sets in the experiments to ensure the reproducibility of results.
[1] Bai, Zhao, et al. "TimeGAE: A Multivariate Time-Series Generation Method via Graph Auto Encoder." International Conference on Database Systems for Advanced Applications. Singapore: Springer Nature Singapore, 2024.
[2] Feng, Shibo, et al. "Latent diffusion transformer for probabilistic time series forecasting." Proceedings of the AAAI Conference on Artificial Intelligence. Vol. 38. No. 11. 2024.

**Questions:**

Please refer to Weaknesses.

---

> ### Author Response · Authors · 2024-11-25
>
> Thank you very much for your comments. Our peer-to-peer responses can be found below. Also, we have uploaded a corrected PDF version of the article.
>
> **answer to w1:**
> Essentially, ours is a research article that provides users with a sound recommendation guide. The proposed CCD is a combination of current techniques to overcome the defect that the central discriminator in the channel-independent generation method cannot handle the small sample time series classification problem. We then conclude that the recommendation priority of the channel-independent method should be improved. We believe that the recommendation guide is valuable to the field and users.
>
> **answer to w2:**
> We conducted experiments based on DLG and Air, which are from the traffic flow dataset and air dataset respectively. The experimental results are as follows:
>
> | Dataset | DLG || | | | |
> |-|-|-|-|-|-|-|
> | Method | CCDGAN | TimeVQVAE | TimeVAE | COSCI-GAN | LS4 | RTSGAN |
> |MDD | 0.284 | 0.293 | 0.301 | 0.241 | 0.227 | 0.238 |
> |ACD | 0.121 | 0.162 | 0.164 | 0.137 | 0.117 | 0.178 |
> |  SD | 0.214 | 0.235 | 0.209 | 0.257 | 0.216 | 0.227 |
> |   KD | 12.341 |12.25 | 12.175 | 12.275 | 12.377 | 12.375 |
> |  ED | 1.12 |1.363 | 1.315 | 1.641 | 1.363 | 1.177 |
> | DTW | 2.346 |2.367 | 2.476 | 2.316 | 2.375 | 2.438 |
> | PS | 0.415 |0.488 | 0.446 | 0.461 | 0.425 | 0.427 |
> |  DS | 0.245 |0.221| 0.237 | 0.257 | 0.227 | 0.288|
>
> | Dataset | Air| | | | | |
> | -| -| -|-|-|-|-|
> | Method |CCDGAN | TimeVQVAE | TimeVAE | COSCI-GAN | LS4 | RTSGAN |
> |  MDD | 0.139 | 0.127 | 0.121 | 0.142 | 0.148 |0.117|
> | ACD | 0.109 | 0.117 | 0.109 | 0.117 | 0.115|0.126 |
> | SD | 0.356 | 0.328 | 0.387 | 0.382 | 0.361 | 0.371 |
> | KD | 8.147 | 8.278 | 8.178 | 8.187 | 8.169 | 8.171 |
> | ED | 0.816 | 0.828 | 0.827 | 0.826 | 0.829 | 0.877 |
> | DTW | 2.044 | 2.28 | 2.091 | 2.081 | 2.062 | 2.027 |
> | PS | 0.404 | 0.483 | 0.426 | 0.433 | 0.429 | 0.416 |
> | DS | 0.107 | 0.186 | 0.131 | 0.124 | 0.139 | 0.121|
>
> **Simulated data**. At the same time, in order to demonstrate the effectiveness of our method, we used the following scheme to generate a simulated time series dataset: we adopt a framework based on the principle Y=f(X). Specifically, the process begins with sampling sequences X from a predefined random mixture distribution.
>
> Next, we employ a binary tree structure to randomly generate mathematical expressions that define the function f(X). This mechanism ensures variability and diversity in the functional relationships represented by f.
>
> Finally, by forward-propagating the sampled sequences X through the generated mathematical expressions f(X), we obtain the corresponding output sequences Y=f(X).
> This methodology provides a flexible and systematic way to simulate complex time-series data with customizable properties, which is critical for evaluating the robustness and generalization capabilities of our proposed models. The details of this dataset can be found in Appendix L.
>
> We generated 100 simulated data sets and conducted experiments according to the experimental design in Figure 3 in the main text. The average results of the experiments are shown below:
>
> | Method| CCDGAN| TimeVQVAE| COSCI-GAN| LS4| Diffusion-TS |
> | -| -| -| -| -| - |
> | MDD$\downarrow$| 0.251| 0.282| 0.272| 0.258| **0.247** |
> | ACD$\downarrow$| 0.116| 0.128| 0.107| **0.105**| 0.114 |
> |  SD$\downarrow$| **0.241**| 0.259| 0.244| 0.248| 0.246 |
> | KD$\downarrow$| **11.01**| 11.017| 11.028| 11.027| 11.014  |
> | ED$\downarrow$| 1.153| 1.151| 1.158| 1.155| **1.151** |
> | DTW$\downarrow$| **2.466**| 2.469| 2.474| 2.471| 2.472 |
> | PS$\downarrow$| 0.164| **0.162**| 0.169| 0.175| 0.166 |
> | DS$\downarrow$| 0.205| 0.217| 0.212| 0.219| **0.202**|
>
> The results demonstrate that both the diffusion-based channel-mixing framework Diffusion-TS and the GAN-based channel-independent framework CCDGAN excel across multiple evaluation metrics, with each achieving top rankings in three categories. CCDGAN outperforms in two distance-based metrics, likely due to GANs' strength in capturing temporal distance characteristics. Meanwhile, Diffusion-TS exhibits balanced performance across all metrics, highlighting the overall effectiveness of diffusion-based generation models.
>
> **answer to w3:**
> We will cite TimeGAE, LDT and SDFormer in the article. At the same time, we also found a new related article SDFormer  published in NIPS2024, which was published after our submission, but no official code was found for TimeGAE and SDFormer. From the results in Table 2 and Table 3 of SDFormer, we can see that the performance of SDFormer is better than DiffusionTS. Combined with our Table 8, we can increase the recommendation priority of SDFormer.
> At the same time, we choose Gamma using grid search in the range of 0.5 to 10. We also performed other hyperparameter analyses (such as kernel size). The specific results are shown in J.1.
>
> **answer to w4:**
> We followed the settings in TSGBench, where the training and test sets were divided using a 9 to 1 ratio.

---

### Official Review · Reviewer_u8EF · 2024-11-04

**Soundness:** 2
**Presentation:** 2
**Contribution:** 2
**Rating:** 5
**Confidence:** 4

**Summary:**

This paper introduces TSGGuide, an updated recommendation guide for multivariate time series generation, aimed at refining the selection of generation methods. Through comprehensive analysis, the authors identified limitations in existing frameworks, specifically the limited exploration of channel-independent frameworks and a lack of evaluation for diffusion-based models. By enhancing the central discriminator within the channel-independent framework, incorporating methods such as TimesNet and attention mechanisms, and introducing CCD, the study demonstrates notable performance improvements in small-sample scenarios.

**Strengths:**

- This work improves upon previous guidelines by prioritizing both channel-independent and diffusion-based generative frameworks, providing users with a more comprehensive and practical TSGGuide for selecting time series generation frameworks.

- By enhancing the central discriminator in the channel-independent framework with advanced techniques (e.g., TimesNet, attention mechanisms, CCD), the study demonstrates clear performance gains in small-sample scenarios, contributing strong methodological advancements.

- The proposed framework shows effective performance in enhancing CCD and CCD-based models, as supported by experimental results.

**Weaknesses:**

- **Motivation:** The motivation for this work is not fully justified. The rationale for addressing limitations in channel-independent methods and diffusion models in MTSG for small-sample datasets seems somewhat forced. Small-sample datasets tend to yield low-quality results in diffusion models due to limited information, challenging the model’s ability to learn data distributions. Despite this, the paper shows that CCDGAN’s performance is comparable to that of Diffusion-TS, suggesting a lack of theoretical and empirical support for this motivation.
- **Dataset Limitation:** Although the paper claims TSGGuide is a highly flexible mechanism capable of adapting to diverse tasks, it only evaluates stock, energy, and medical datasets. The limited selection constrains the demonstration of the module’s plug-and-play capability, which may impact the credibility of its claimed flexibility across various domains.
- **Channel-Independent Approach:** The paper’s choice to adopt a channel-independent framework raises questions when applied to stock datasets, which exhibit strong inter-channel correlations (e.g., stocks in the same sector often follow similar trends). While the model demonstrates effectiveness, it may overlook inherent characteristics of financial data by disregarding these correlations.

**Questions:**

- Given that only stock, energy, and medical datasets were used, how might this affect the generalizability and credibility of TSGGuide’s flexibility across different domains?

- In small-sample contexts, GANs generally outperform diffusion-based models. Yet, as noted in the paper, CCDGAN does not significantly outperform Diffusion-TS. Can the authors provide additional evidence to substantiate their theoretical claims?

- Is CCDGAN essentially a combination of the CCD mechanism with GAN? The methodology section lacks detail here; could you provide more on how CCDGAN was developed?

- Given the emphasis on a channel-independent framework, why are there no comparisons with channel-mixed framework methods? Since channel-independent frameworks ignore inter-sequence interactions, empirical evidence is needed to validate their benefits, especially for financial datasets where sequences are highly correlated.

---

> ### Author Response · Authors · 2024-11-25
>
> Thank you very much for your comments. Our peer-to-peer responses can be found below. Also, we have uploaded a corrected PDF version of the article.
>
> **answer to w1:**
> Our primary motivation was not to address the limitations of diffusion models on small-sample datasets but rather to achieve two main objectives: (1) to analyze existing time series generation methods and establish comprehensive recommendation guidelines, enabling users to select appropriate models for diverse scenarios; and (2) to enhance the effectiveness of channel-independent methods by designing the Central Convolution Discriminator (CCD), which significantly improves their recommended priority. Experimental results strongly support both objectives.
>
> At the same time, we also compared the performance of CCDGAN and Diffusion-TS on a small sample data set. We used the Stock data set and randomly sampled 10% of the data. The results are as follows:
>
> |  Dataset | 10\% Stock | - |
> |  - | - |-  |
> | Method | CCDGAN | Diiffusion-TS |
> | MDD | 0.511 | 0.613 |
> |  ACD | 0.049 | 0.058 |
> | SD | 0.2 | 0.244 |
> | KD | 0.861 | 1.015 |
> |  ED | 2.239 | 1.996 |
> | DTW | 5.717 | 5.931|
>
> Among the six evaluation metrics, CCDGAN outperforms DiffusionTS on four, demonstrating its superiority on small datasets. This advantage arises from CCDGAN’s channel-independent design and the Central Convolution Discriminator (CCD), which are better suited for capturing sparse data distributions. In contrast, diffusion models like DiffusionTS rely on larger datasets for effective training, limiting their performance in such scenarios.
>
> **answer to w2:**
> We conducted experiments based on DLG and Air, which are from the traffic flow dataset and air dataset respectively. The experimental results are as follows:
>
> | Dataset | DLG |   |   |   |   |  |
> | --- | --- | --- | --- | --- | --- | --- |
> | Method | CCDGAN | TimeVQVAE | TimeVAE | COSCI-GAN | LS4 | RTSGAN |
> |  MDD | 0.284 | 0.293 | 0.301 | 0.241 | 0.227 | 0.238 |
> |  ACD | 0.121 | 0.162 | 0.164 | 0.137 | 0.117 | 0.178 |
> |  SD | 0.214 | 0.235 | 0.209 | 0.257 | 0.216 | 0.227 |
> |   KD | 12.341 | 12.25 | 12.175 | 12.275 | 12.377 | 12.375 |
> |  ED | 1.12 | 1.363 | 1.315 | 1.641 | 1.363 | 1.177 |
> | DTW | 2.346 | 2.367 | 2.476 | 2.316 | 2.375 | 2.438 |
> | PS | 0.415 | 0.488 | 0.446 | 0.461 | 0.425 | 0.427 |
> |  DS | 0.245 | 0.221 | 0.237 | 0.257 | 0.227 | 0.288|
>
> | Dataset | Air|   |   |   |   |  |
> | --- | --- | --- | --- | --- | --- | --- |
> | Method | CCDGAN | TimeVQVAE | TimeVAE | COSCI-GAN | LS4 | RTSGAN |
> |  MDD | 0.139 | 0.127 | 0.121 | 0.142 | 0.148 | 0.117 |
> | ACD | 0.109 | 0.117 | 0.109 | 0.117 | 0.115 | 0.126 |
> | SD | 0.356 | 0.328 | 0.387 | 0.382 | 0.361 | 0.371 |
> | KD | 8.147 | 8.278 | 8.178 | 8.187 | 8.169 | 8.171 |
> | ED | 0.816 | 0.828 | 0.827 | 0.826 | 0.829 | 0.877 |
> | DTW | 2.044 | 2.28 | 2.091 | 2.081 | 2.062 | 2.027 |
> | PS | 0.404 | 0.483 | 0.426 | 0.433 | 0.429 | 0.416 |
> | DS | 0.107 | 0.186 | 0.131 | 0.124 | 0.139 | 0.121|
>
> **Simulated data**. At the same time, in order to demonstrate the effectiveness of our method, we used the following scheme to generate a simulated time series dataset: we adopt a framework based on the principle Y=f(X). Specifically, the process begins with sampling sequences X from a predefined random mixture distribution.
>
> Next, we employ a binary tree structure to randomly generate mathematical expressions that define the function f(X). This mechanism ensures variability and diversity in the functional relationships represented by f.
>
> Finally, by forward-propagating the sampled sequences X through the generated mathematical expressions f(X), we obtain the corresponding output sequences Y=f(X).
> This methodology provides a flexible and systematic way to simulate complex time-series data with customizable properties, which is critical for evaluating the robustness and generalization capabilities of our proposed models. The details of this dataset can be found in Appendix L.
>
> We generated 100 simulated data sets and conducted experiments according to the experimental design in Figure 3 in the main text. The average results of the experiments are shown below:
>
> | Method| CCDGAN| TimeVQVAE| COSCI-GAN| LS4| Diffusion-TS |
> | -| -| -| -| -| - |
> | MDD$\downarrow$| 0.251| 0.282| 0.272| 0.258| **0.247** |
> | ACD$\downarrow$| 0.116| 0.128| 0.107| **0.105**| 0.114 |
> |  SD$\downarrow$| **0.241**| 0.259| 0.244| 0.248| 0.246 |
> | KD$\downarrow$| **11.01**| 11.017| 11.028| 11.027| 11.014  |
> | ED$\downarrow$| 1.153| 1.151| 1.158| 1.155| **1.151** |
> | DTW$\downarrow$| **2.466**| 2.469| 2.474| 2.471| 2.472 |
> | PS$\downarrow$| 0.164| **0.162**| 0.169| 0.175| 0.166 |
> | DS$\downarrow$| 0.205| 0.217| 0.212| 0.219| **0.202**|

---

> > ### Author Response · Authors · 2024-11-25
> >
> > The results demonstrate that both the diffusion-based channel-mixing framework Diffusion-TS and the GAN-based channel-independent framework CCDGAN excel across multiple evaluation metrics, with each achieving top rankings in three categories. CCDGAN outperforms in two distance-based metrics, likely due to GANs' strength in capturing temporal distance characteristics. Meanwhile, Diffusion-TS exhibits balanced performance across all metrics, highlighting the overall effectiveness of diffusion-based generation models.
> >
> > **answer to w3:**
> > First, we can use convolution to capture the correlation between channels through the mechanism (as shown in Formula 3). At the same time, we show the advantage of CCD in capturing correlation in Table 11.
> >
> > **answer to q1:**
> > see answer to w2 .
> >
> > **answer to q2:** We are exploring the performance of different methods in small sample environments and forming corresponding recommendation guidelines. As shown in our experimental results, different methods have different advantages in different scenarios. We do not emphasize that CCDGAN is definitely better than the diffusion model.
> >
> > **answer to q3:**
> > As shown in Fig. 1, this is a basic architecture of CCDGAN, the time series generation framework uses LSTM and CCD is used to replace the central discriminator, the loss function and model training are shown in Appendix A.
> >
> > **answer to q4:**
> > As shown in Figure 1, the channel-independent framework leverages a central discriminator to capture interactions between sequences. It is important to note that the channel-independent framework does not generate each channel independently; the term is used merely to distinguish it from channel-mixing frameworks. The detailed architecture can be found in Section 3.3. Except for CCD-GAN and COSCI-GAN, all other models are channel-mixing generation models.

---

### Author Response · Authors · 2024-11-25

Thank you for your insightful feedback. In response, we have revised the manuscript and uploaded the updated PDF, with changes highlighted in blue. Key revisions include:

**Clarification of Motivation:** To enhance clarity, we emphasize that the primary objective of this work is to analyze current time series generation methods and develop an effective recommendation framework, providing users with informed guidance across various scenarios. To align with this goal, we have moved the validation of the Central Convolution Discriminator (CCD) to the supplementary materials, ensuring it does not overshadow the core motivation.

**Increased Dataset Diversity:** To improve the robustness and accuracy of the proposed guidelines, we have incorporated datasets from diverse domains, including DLG (traffic data) and Air. Additionally, we generated 100 synthetic datasets using the methodology of Meidani et al. (2023) for comprehensive evaluation.

---

### Author Response · Authors · 2024-11-28

Dear Reviewers,

As the author-reviewer discussion period nears its deadline, we kindly ask you to review our responses to your comments, concerns, and suggestions. If you have any additional questions or feedback, we will do our best to address them promptly before the discussion period concludes. If our responses have satisfactorily addressed your concerns, we would greatly appreciate it if you could update your evaluation of our work accordingly.

Thank you once again for your valuable time and thoughtful feedback.

Sincerely,

Authors

---

### Meta-Review · Area_Chair_zDX6 · 2024-12-18

**Metareview:**

The paper proposes a recommendation guide for MTSG methods, incorporating diffusion-based techniques and a channel-independent framework. The paper’s strengths include a comprehensive analysis, clear explanations, and improved guidelines for small-sample scenarios. However, it suffers from limited dataset diversity, insufficient exploration of channel-independent frameworks, and a lack of technical novelty.


Despite the authors’ rebuttal, the paper’s weaknesses, such as unclear equations, limited datasets, and insufficient originality, remain. The reviewers appreciated the clear language and comprehensive experiments but found the main contribution unclear and the writing confusing. Given these issues, I recommend rejecting the paper as it does not meet the necessary standards for acceptance.

**Additional Comments On Reviewer Discussion:**

The authors responded by clarifying motivations, adding dataset diversity, and providing detailed experimental results, but these efforts did not fully address the reviewers’ concerns.

---

### Decision · Program_Chairs · 2025-01-22

Reject